# Genome-wide analyses of LATERAL ORGAN BOUNDARIES in cassava reveal the role of LBD47 in defence against bacterial blight

Yiming Mao[1,Ⓔ], Assane Hamidou Abdoulaye[1,Ⓔ], Jiming Song[2], Xiaowen Yao[1], Yijie Zhang[1], Yu Gao[1], Yiwei Ye[1], Kai Luo[1], Wei Xia[1]*, Yinhua Chen[1]*

1 Hainan Key Laboratory for Sustainable Utilisation of Tropical Bioresources, College of Tropical Crops, Hainan University, Haikou, Hainan, China, 2 Institute of Tropical and Subtropical Cash Crop, Yunnan Academy of Agricultural Sciences, Kunming, China

Ⓔ These authors contributed equally to this work.
* saizjxiawei@hainanu.edu.cn (WX); yhchen@hainanu.edu.cn (YC)

## Abstract

The *Arabidopsis thaliana* ASYMMETRIC LEAVES2 (*AS2*) gene is responsible for the development of flat, symmetric, and extended leaf laminae and their veins. The *AS2* gene belongs to the plant-specific *AS2-LIKE/LATERAL ORGAN BOUNDARIES* (*LOB*)-domain (*ASL/LBD*), which consists of 42 proteins in *Arabidopsis* with a conserved amino-terminal domain known as the *AS2/LOB* domain, and a variable carboxyl-terminal region. *AS2/LOB* domain consists of an amino-terminal (N-terminal) that contains a cysteine repeat (the C-motif), a conserved glycine residue, and a leucine-zipper-like. *AS2/LOB* domain has been characterised in plants such as *A. thaliana*, *Zea mays*, and *Oryza sativum*. Nevertheless, it remains uncharacterised in cassava (*Manihot esculenta*). Characterisation and identification of cassava *ASL/LBD* genes using the computational algorithms, hidden Markov model profiles (PF03195), determined 55 *ASL/LBD* genes (*MeASLBD1* to *MeASLBD55*). The gene structure and motif composition were conserved in *MeASLBDs*, while the expression profiles of these genes were highly diverse, implying that they are associated with diverse functions. Weighted gene co-expression network analysis (WGCNA) of target genes and promoter analysis suggest that these *MeASLBDs* may be involved in hormone and stress responses. Furthermore, the analysis of cis-regulatory elements in promoter regions suggested that *MeASLBDs* may be involved in the plant phytohormone signal response. The transcriptome data of cassava under biotic and abiotic stresses revealed that *MeASLBD46* and *MeASLBD47* greatly respond to disease and drought. The *MeASLBD47* gene was selected for functional analysis. The result indicated that *MeASLBD47* significantly mitigated the virulence of cassava bacterial blight (*Xam*CHN11) through Real-Time Quantitative Reverse Transcription PCR (qRT-PCR) and Virus-induced gene silencing (VIGS). These findings provided a comprehensive analysis of *ASL/LBD* genes and laid the groundwork for future research to understand *ASL/LBD* genes.

**Data Availability Statement:** All relevant data are within the manuscript and its Supporting Information files.

**Funding:** This study was supported by National Key Research and Development Program (2018YFD1000500), the Natural Science Foundation of Hainan Province (320QN193) and China Agriculture Research System (CARS-11-hncyh). National Key Research and Development Program (2018YFD1000500) Funding had role in study design and material collection; the Natural Science Foundation of Hainan Province (320QN193) had role in the research of qPCR data collection and analysis; China Agriculture Research System (CARS-11-hncyh) founders had role in RNA-Seq data collection and analysis, manuscript preparation.

**Competing interests:** The authors have declared that no competing interests exist.

## Introduction

Organ development in multicellular systems is governed by the activation of a new genetic program and the repression of a previously active program, which is largely governed by epigenetic systems. The prime model for studying such positive and negative control programs is the *Arabidopsis thaliana* leaf developmental process with an adaxial-abaxial (dorsal-ventral) polarity that is epigenetically regulated by the repressor complex *ASYMMETRIC LEAVES1* (*AS1*)-*AS2* [1–4]. *AS2* gene has received the most extensive investigation in genetic and molecular research [5]. The *AS2* gene in *Arabidopsis* is a key regulator in the development of flat symmetric leaves with vascular bundles and fine networks of venation systems, the morphology of which appears to have evolved suitably for efficient photosynthesis [5]. For instance, *Arabidopsis AS2* regulates leaf adaxial-abaxial partitioning by suppressing the expression of the abaxial-determining gene *ETTIN/AUXIN RESPONSE FACTOR3* (*ETT/ARF3*), implying that *AS2* is involved in epigenetic repression of *ETT/ARF3* by gene body DNA methylation [6]. *AS2* specifically binds the CpG-containing sequence in exon 1 of *ETT/ARF3*, and the binding requires the zinc-finger-like motif in *AS2*, which is structurally similar to the zinc-finger-CxxC domain in vertebrate DNA methyltransferase1 [6]. In addition, AS2 is involved in epigenetic repression of the abaxial *ARF4* and class 1 *KNOX* homeobox genes by forming a complex with the MYB protein. The cooperative action of various modifier genes significantly boosts the repressed expression of these genes by AS2 [6].

Transcription factor (TF) families play critical roles in plant growth, development, and environmental stress responses [7]. The *ASYMMETRIC LEAVES2-LIKE/LATERAL ORGAN BOUNDARIES DOMAIN* (*ASL/LBD*) family is an important plant-specific TF family [8]. These members are also named the <u>L</u>ateral <u>O</u>rgan <u>B</u>oundary (*LOB*) <u>D</u>omain (*LBD*). *ASL/LBDs* are defined by a highly conserved *LOB* domain of approximately 100 amino acids (aa) [9]. *AS2* gene belongs to the plant-specific *AS2/LOB* protein family, which contains 42 members in *Arabidopsis* with a conserved amino-terminal domain known as the *AS2/LOB* domain, and a variable carboxyl-terminal region; however, little is known about their functions [9–13]. The *AS2/LOB* domain contains (i) a zinc-finger-like motif ($CX_2CX_6CX_3C$) required for binding specific DNA sequences (5′ GCGGCG 3′), and the interaction with basic helix-loop-helix (bHLH) proteins could reduce this affinity; (ii) an invariant glycine residue, Gly-Ala-Ser (GAS), critical for the biological function of *AS2/LOB* proteins in *Arabidopsis*; and (iii) a leucine-zipper-like (LZL-region; $LX_6LX_3LX_6L$) which may play a role in the protein-protein interactions to form homo and/or hetero multimers [9–11, 14]. Importantly, the crystal structure of the homodimeric *LOB* domain of Ramosa2 from wheat (TtRa2LD) was crystallized and determined [15]. The structure mainly consists of a zinc finger, a GAS motif consisting of two α-helices, a highly conserved five-residue motif (Asp-Pro-Val-Tyr-Gly, known as DPVYG motif), and an amphipathic α-helix with the feature of leucine zipper-like coiled-coil element. According to biochemical, molecular modeling, and small-angle X-ray scattering analysis, dimerization is important for cooperative DNA binding and palindromic DNA discrimination through a molecular calipers mechanism [15]. Based on the presence of the LZL-region, the *ASL/LBD* gene family can be classified at least into two classes: "Class I" contains all these conserved domain motifs, whereas "Class II" contains only structural motifs similar to the zinc-finger-like motif [9–11, 16]. "Class I" is further divided into two subclasses (Ia and Ib) [5]. A base substitution mutation in the conserved glycine codon of *ASL/LBD*5 causes the typical phenotypic alterations observed in numerous *ASL/LBD* mutants, making this residue essential for *ASL/LBD* function [11, 14]. In addition, the *ASL/LBD*5 mutant, which contains a mutation in the region encoding the carboxy-terminal (C-terminal) half of *ASL/LBD*, results in the typical mutant phenotype; thus, the C-terminal half also plays a role in *ASL/LBD* function [17].

The first *ASL/LBD* gene (AT5G63090) was identified in *A. thaliana* [9]. Since then, *ASL/LBD* genes have been continually reported in various plants and associated with diverse functions (S1 Table in S1 File). For instance, *AS2* (*AS2/LOB6*) in *Arabidopsis* is involved in a regulatory loop that maintains the shoot meristem and controls leaf polarity and flower development by interacting with an MYB motif encoded by *AS1*, recently defined as the SANT domain [12, 14, 18]. In addition, it was found that the loss-of-function of *AS2/LOB20* in *Arabidopsis* enhanced resistance to the root-infecting vascular wilt pathogen *Fusarium oxysporum* [19]. Furthermore, the expression profiles of *A. thaliana* (*At*) *LBD* members indicated that pathogen inoculation induced the expression of *AS2/LOB37* and *AS2/LOB38* [19]. *CsLOB1*, a homolog of *AS2/LOB1* and *AS2/LOB11*, acts as a target of transcription activator-like (TAL) effectors following infection with bacterial canker disease [20]. These findings imply that the *ASL/LBD* genes play important roles in plant defence responses.

Cassava (2n = 36; *Manihot esculenta Crantz*) is a major staple crop in tropical regions and the third most consumed grain after rice and maize [21–23]. *M. esculenta Crantz* originated from its wild ancestor, *M. esculenta* ssp. *Flabellifolia* [24]. The cassava root crop provides staple food for over 700 million people worldwide [23, 25]. Cassava is highly drought-resistant, and its storage roots can be preserved in the soil for a few years, making it an essential carbohydrate source to alleviate global famine [26]. Besides, it is an ideal feedstock crop for bioenergy, bio-materials, and animal feeds due to its advantageous agricultural characteristics and high starch quantity and quality [27, 28]. Nonetheless, cassava production is constrained by several plant pathogens that threaten the food security of millions of people worldwide [29–31]. Cassava's most serious bacterial disease is Cassava Bacterial Blight (CBB), caused by *Xanthomonas axonopodis pv. manihotis* (*Xam*) [32]. CBB threatens food security in tropical regions and can generate up to 100 percent losses under favourable climatic conditions (CABI, 2015; FAO, 2008) [33]. The rapid spread of CBB in some cassava-producing regions and the emergence of new disease reports in regions where cassava is a staple crop highlight the necessity of developing novel methods to control this plant disease [30, 34–36]. Cassava genome publication lays the groundwork for genome-wide analysis of new gene resources [37]. Nonetheless, studies on the molecular mechanisms of CBB resistance are scarce [38]. The characteristics underlying the biotic stress response of cassava remain largely unknown. This investigation was conducted to identify the *ASL/LBD* gene family in cassava and determine LOB expression profil*e*s under pathogen stress. Identifying and characterising the *ASL/LBD* genes and determining their expression profiles under biotic stress could provide valuable insight for plant disease control.

This study identified 55 *M. esculenta ASL/LBDs* (*MeASLBDs*) from the cassava genome. The phylogeny, conserved motifs, gene structures, and expression profiles of *MeASLBDs* were thoroughly determined. Moreover, potential target genes were identified *via* promoter analysis, and a co-expression network was constructed along with *MeASLBDs* based on a transcriptome dataset derived from pathogen treatment. The *MeASLBD47* gene expression was highly induced following inoculation with cassava bacterial blight pathogen (*Xam*CHN11) and was selected for functional analysis. The results indicated that *MeASLBD47* significantly mitigated the virulence of cassava bacterial blight (*Xam*CHN11) through qRT-PCR and Virus-induced gene silencing (VIGS). These findings provide invaluable insight for further physiological and functional studies of the *AS2/LOB* domain in cassava and highlight the possible functions in pathogen response.

## Materials and methods

### Plant material and growth conditions

The cassava cultivar used in this study were SC8, Arg7, W14, and KU50. The stems were cut into three-node sections and grown in a pest-free sterile phytotron at temperatures ranging from 35°C to 25°C under a 16/8-h photoperiod and 80% relative humidity. The leaves (30-day-

old) were inoculated with *Xanthomonas axonopodis* pv. *Manihotis* (*Xam*CHN11) [32]. The leaves of five cassava plants (nine leaves per plant) were inoculated by dropping 10-μL of *Xam*CHN11 suspension of $1 \times 10^8$ CFU/mL [optical density at 600 nm ($OD_{600}$) = 0.1] into a 2-mm-diameter ring. Cassava seedlings were sampled at 1 h, 3 h, 6 h, and 9 h post-inoculation. All samples were frozen in liquid nitrogen and stored at -80°C until use. To quantify the size of the lesion, typical *Xam* symptoms, such as necrosis and chlorosis around the inoculation point, were considered. At 3 days post-inoculation (dpi), lesions were measured on eight leaves per treatment and then plotted. The experiments were conducted at least twice with identical results.

## RNA isolation, qRT-PCR analysis, and statistics analysis

Total RNA was isolated with the RNAprep Pure Plant Kit (DP441, TIANGEN, Beijing, China). RNA concentration determination, DNase I, RNase-free (1 U/μL) treatment, cDNA synthesis, qRT-PCRs, and data analysis were performed following published protocols with minor modifications [39, 40]. Photometric UV-Vis RNA/DNA quantification was performed using NanoDrop One/One$^C$. The cDNA was synthesized with the FastKing RT Kit (TIANGEN, Beijing, China). The gene-specific primers were designed with the Primer 3.0 program (S2 Table in S1 File). qRT-PCR was performed in a reaction system of 20 μL containing 10 μL TB Green Premix Ex Taq II, 1 μL 10 μM forward primer, 1 μL 10 μM reverse primer, 2 μL cDNA, and 6 μL ddH2O. The PCR amplification conditions were set as follows: denaturation at 95°C for 30 s, followed by 40 cycles of 95°C for 5 s, 60°C for 35 s, 95°C for 15 s, and 60°C for 1 min. The *EF1α* gene was selected to calculate the relative fold differences using the $2^{-\Delta\Delta CT}$ method [$\Delta\Delta C_T = (Ct_{target\ gene} - Ct_{EF1\alpha})$] [40]. Statistics analysis was conducted using SPSS software. Rstudio software was used to infer the heatmap. Three sets of biological replicates and three sets of technical replicates were analysed.

## Generation of pCsCMV VIGS construct and agroinfiltration

VIGS tool (https://vigs.solgenomics.net/) was applied to analyse the CDS sequence of the gene and obtain the optimal silencing fragment for the construction of VIGS vector [41, 42]. 487 bp cassava *PDS* (Manes.05G193700) and 237 bp *MeASLBD47* (Manes.12G110600) fragments were amplified using cassava cDNAs as a template and the corresponding primer pairs (S2 Table in S1 File). Subsequently, the amplified fragments were cloned into pCsCMV vector to generate pCsCMV-*PDS* and pCsCMV-*AS2/LOB47* using Nimble Cloning [43]. pCsCMV-NC was utilised as a negative control (NC). The pCsCMV-NC-based constructs were transformed into *Agrobacterium tumefaciens* (strain GV3101). A single colony of *A. tumefaciens* (GV3101) for each strain was incubated for 20–36 h at 28°C in Luria–Bertani (LB) medium (10 mL) containing 25 mg/mL kanamycin and 25 mg/mL rifampicin. The bacterial cultures were then centrifuged at 4,000g for 5 min and were resuspended in agroinfiltration buffer [10 mM MgCl2, 10 mM 2-(N-Morpholino) ethanesulfonic acid, and 100 μM acetosyringone] for reaching an optical density of 1.0 at 600 nm ($OD_{600}$) [44]. The agrobacterium suspension was kept at room temperature for 3 h in the dark before inoculation. 2-week old cassava plants were used for agroinfiltration using a 1-mL needleless syringe. The back sides of the healthy and fully developed leaves from the middle of each plant were selected for agroinfiltration. Inoculations were administered at 8–10 spots on both sides of the main vein per leaf to increase the infiltrated leaf area.

## Identification and sequence analysis of *MeASLBD* genes

The cassava genome sequence was downloaded from Phytozome (http://www.phytozome.net) [45]. The *AS2/LOB* protein sequences of *A. thaliana* were retrieved from TAIR 12.0 (www.arabidopsis.org) [46]. The hidden Markov model (HMM) -based profile (Pfam: PF03195) of the putative *AS2/LOB* domain was built using the HMMER program (http://hmmer.org/) [47,

48]. *AS2/LOB* domain obtained from the Pfam database (http://pfam.xfam.org/) [49] was used as a query to identify all potential *AS2/LOB* protein sequences using a BLASTp search E-value threshold of $1.0 \times 10^{-10}$. The putative *AS2/LOB* domain of the candidate sequences was further confirmed by the National Center for Biotechnology Information (NCBI) database (http://www.ncbi.nlm.nih.gov/) and the SMART database (http://smart.embl-heidelberg.de/). The MW and pI of cassava *AS2/LOB* proteins (*MeASLBDs*) were calculated with the online program ExPASy (https://www.expasy.org/) [50].

## Phylogenetic tree construction, conserved motifs, and structure analysis

To determine the evolutionary relationships between *M. esculenta* and *A. thaliana ASL/LBD* gene family, phylogenetic analysis was performed with MEGA 7.0 by constructing a minimum evolution (ME) phylogenetic tree using substitution models G + I with 1,000 bootstrap replicates to support statistical reliability [51]. The full-length amino acid sequences of *M. esculenta* and *A. thaliana AS2/LOB* protein sequences were aligned using the ClustalX program with the default parameters. The results were displayed with DNAMAN 9.0 (Lynnon Biosoft) software. The Multiple Em for Motif Elucidation (MEME, version 5.4.1.) tool (http://meme-suite.org/tools/meme) was used to predict the conserved motifs of the *MeASLBD* protein sequences [52]. The relative parameters were set to an optimum motif width of 25–50 and a maximum number of 10 motifs. The exon-intron structure of *MeASLBD* genes was analysed using the Gene Structure Display Server (http://gsds.gao-lab.org/) [53].

## Chromosomal location, gene duplication event, gene collinearity, and synteny analysis

The chromosomal location information of 55 *MeASLBD* genes was extracted from the cassava genomic annotation file GFF3 (general feature format), retrieved from Ensembl Plants (http://plants.ensembl.org/index.html). The chromosomal distribution of *MeASLBD* genes was visualised using TBtools software [54]. Duplication detection for *MeASLBD* genes was performed using the Multiple Collinearity Scan Toolkit (MCScanX) with an E-value of $10^{-5}$ [55]. The homologous regions of the *MeASLBD* genes were identified with the MCScanX (http://chibba.pgml.uga.edu/mcscan2/) (https://github.com/wyp1125/MCScanX) program to determine the level of synteny [55]. Tandem and segmental duplication events were identified and visualised with Circos software (http://circos.ca/) version 0.69 [56]. The $K_a/K_s$ value was calculated as the ratio of the number of nonsynonymous substitutions per nonsynonymous site ($K_a$) to the number of synonymous substitutions per synonymous site ($K_s$) over a given period.

## Cis-acting element analysis and identification of *MeASLBDs* target gene

The promoter regions of 55 *MeASLBD* genes, the upstream (2000 bp) of the translation initiation codons, were extracted from the Phytozome database of the cassava genome (https://phytozome.jgi.doe.gov/). These sequences were applied in PlantCARE database (https://bioinformatics.psb.ugent.be/webtools/plantcare/html) [57] to predict the cis-acting elements present in the promoter regions of the *MeASLBDs*, and visualised with TBtools. The genes containing the motif (5′ GCGGCG 3′) in the upstream promoter region are candidate target genes for *MeASLBD*. Blast2GO tool was used to annotate the candidate target genes [58].

## Gene expression profiles of *MeASLBDs* in cassava

The expression profiles of 55 *MeASLBD* genes in root and leaf tissues of cassava cultivars Argentina 7 (Arg7), wild subspecies 14 (W14), and Kasetsart University 50 (KU50) were

analysed. RNA-seq data of the Arg7, W14, and KU50 were retrieved from NCBI database (https://www.ncbi.nlm.nih.gov/geo/query/acc.cgi?acc=GSE93098) (S7 Table in S1 File). HISAT2 (https://github.com/DaehwanKimLab/hisat2) software [59] was applied with default parameters to map the reads. Under default parameters, the StringTie program [60] was applied to assemble transcripts and compute reads per kilobase million (RPKM) values. The cassava *AS2/LOB* family heatmaps were established with log2-transformed RPKM values and visualised with Tbtools [54]. The min-max method was used to normalise the RPKM values.

### Co-expression analysis of *MeASLBDs* and identification of *MeASLBD* interacting genes by WGCNA

A co-expression analysis of *MeASLBD* and its target genes was generated using 55 selected transcriptomes (S3 Table in S1 File). The RPKM values of *MeASLBDs* and their predicted target genes were screened, and genes with an RPKM value $< 1$ were removed. Weighted Gene Co-expression Network Analysis (WGCNA) was performed using the Rstudio environment to identify the selected co-expression module [61]. The selected co-expression module is considered a cluster of highly related genes. The module genes showing expression patterns consistent with *MeASLBDs* were selected for Cytoscape visual analysis [62].

## Results

### Identification and chromosome localisation of cassava *ASL/LBD* genes

55 *ASL/LBD* genes were identified in cassava and named *MeASLBD* (*MeASLBD1* to *MeASLBD55*). The predicted protein products of *MeASLBDs* ranged from 116 to 429 aa, with the molecular weight (MW) varying from 13.14 (*MeASLBD28*) to 46.78 kDa (*MeASLBD55*). The isoelectric point (pI) of their proteins varied from 4.67 (*MeASLBD38*) to 9.02 (*MeASLBD16*). Through chromosomal localisation analysis, 54 *MeASLBD* genes were unevenly distributed on the 18 cassava chromosomes (S1 Fig in S1 File). However, *MeASLBD19* was localized on an unanchored scaffold004600 (S1 Fig in S1 File). The largest number of *MeASLBD* genes was found on chromosome 7 (7, 12.7%), followed by chromosomes 12 (6, 10.9%), 5 (5, 9%), 10 (5, 9%), 14 (5, 9%), 6 (4, 7.2%), and 13 (4, 7.2%). The remaining chromosomes contained one or two *MeASLBD* genes.

### Evolutionary relationship and gene structure analysis of the *MeASLBD* genes

To investigate the evolutionary relationship of the *MeASLBD* genes, 43 *AS2/LOB* proteins of *A. thaliana* (*AtASL/LBD*) and 55 *MeASLBD* were used to construct a ME tree (Fig 1) *MeASLBDs* were classified into two major classes: Class I (46) and Class II (9). Class I *ASL/LBD* genes contain a conserved $CX_2CX_6CX_3C$ zinc-finger-like motif and an $LX_6LX_3LX_6L$ LZL-region, while Class II *ASL/LBD* genes contain only a conserved zinc-finger-like motif. Forty-six *MeASLBDs* (83%) were clustered into Class I. Class I was subdivided into seven subgroups (Class Ia to Class Ig). Class Ia contained the highest number of *MeASLBD* (13), followed by 11, 11, 6, 2, 2, and 1 *MeASLBDs* belonging to Class Ib, Ic, Ie, Id, and If, respectively. Nine *MeASLBDs* belonged to Class II, subdivided into two distinct subgroups: Class IIa (4) and Class IIb (5). Interestingly, this result is consistent with *Arabidopsis ASL/LBD* genes classification [9]. Previous studies showed that many *AtASL/LBDs* genes (Class I) were involved in lateral organ development [10]. Class II *ASL/LBD* genes are related to metabolism [10]. The phylogenetic analysis of *A. thaliana* and *M. esculenta* indicated that *MeASLBD* genes might have similar biological functions as *AtASL/LBDs*.

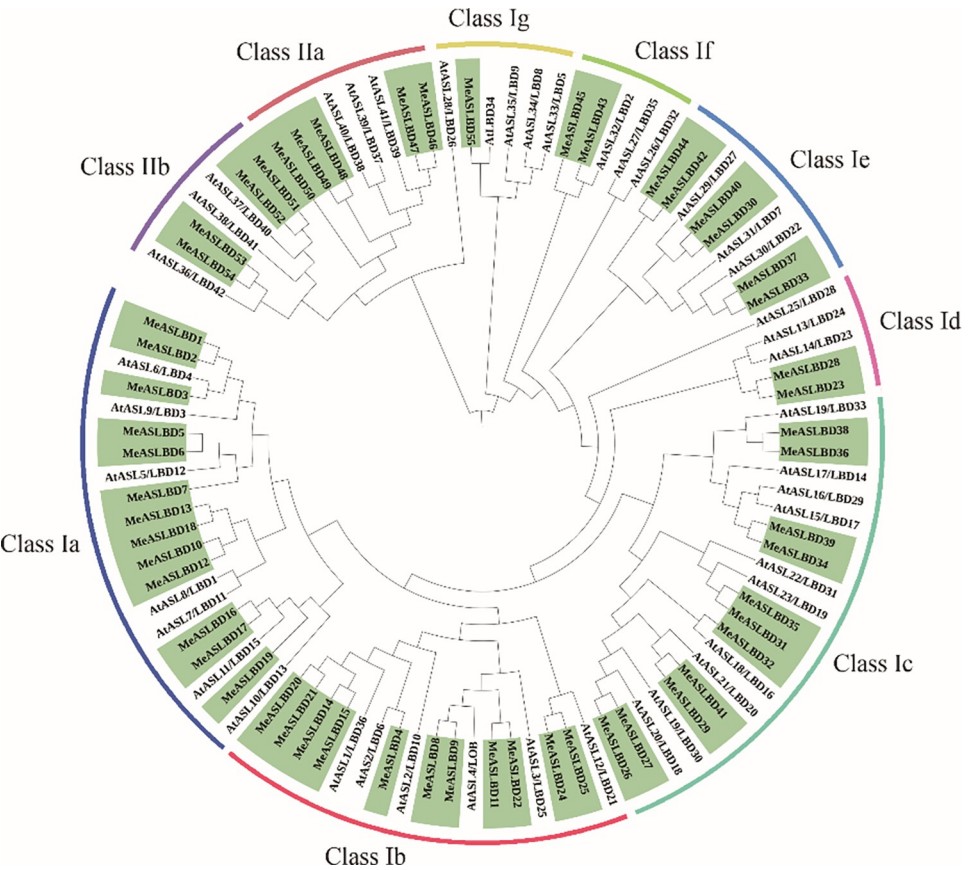

**Fig 1. Phylogenetic analysis of *AS2/LOB* proteins in *M. esculenta* and *A. thaliana*.** The proteins *AtASL/LBD* and *MeASLBD* sequences were used to construct a minimum evolution (ME) tree. The tree was divided into two classes (Class I and II) and nine subclasses (Ia-g and IIa-b).

Gene structure analysis (Fig 2A) showed that *MeASLBD* genes have a simple structure with 2 to 4 introns and 1 to 4 exons. The majority (45) of the *MeASLBD* (82%) genes contains two exons, three *MeASLBD* genes (5%) contain one exon, three *MeASLBD* genes (5%) contain four exons and one *MeASLBD* gene (8%), clustered in Class Ig, contains five exons (Fig 2C). 40 *MeASLBD* coding regions contained a splicing site in the *AS2/LOB* domain (Fig 2C). 11 *MeASLBD* noncoding regions (UTR) contained an intron (Fig 2C). Ten putative motifs were identified using the MEME tool (Fig 2B). The conserved motifs of the *MeASLBD* protein sequences ranged from 25 to 50 aa (motifs 8 and 1, respectively) in length and contained between one (*MeASLBD55*) to six (*MeASLBD14*, *MeASLBD15*, *MeASLBD20* and *MeASLBD21*) motifs. Motifs 1 and 2 were conserved in all *MeASLBD* protein sequences except *MeASLBD55* which contained only motif 2. Cluster analysis showed that Class Ia, Id, Ig, and II have each similar gene structures. Motifs 3, 5, 6, and 10 were only conserved in *MeASLD* proteins of Class I, whereas motifs 8 and 4 were only conserved in *MeASLDs* proteins of Class II, suggesting that they are evolutionary divergent. Notably, motif 2 is conserved in all *MeASLBD* proteins. The *MeASLBD* proteins with similar motif compositions and gene lengths indicate they may share similar functions.

## Conserved motif analysis of *MeASLBD* genes

Multiple sequence alignment of the MeASLBD protein sequences using ClustalX showed that all sequences contained a zinc-finger-like motif ($CX_2CX_6CX_3C$) at the N-terminus (Fig 3). The

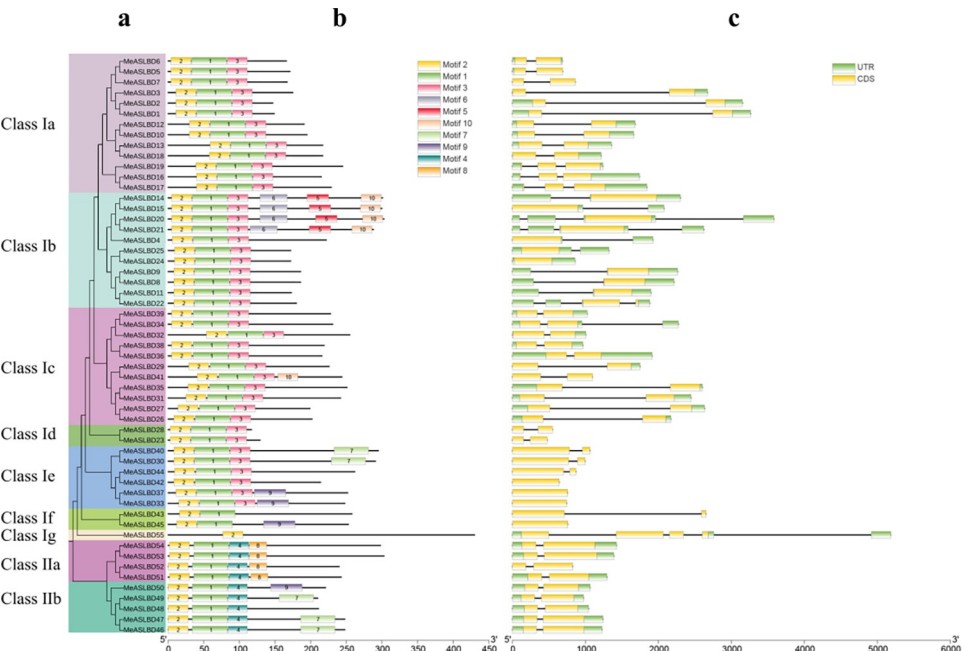

**Fig 2. Phylogenetic analysis of *ASL/LBD* genes and the structures and relative motif positions of the *ASL/LBD* gene family members in *M. esculenta*.** (a) Minimum evolution (ME) tree of *AS2/LOB* proteins in *M. esculenta*. The phylogenetic tree was constructed using MEGA 7.0 software with the bootstrap method (1000 replicates). Branches of the different classes are shown in different colours. (b) Conserved motifs of *MeASLBD* proteins predicted by MEME. (c) Exon-intron structures of the *MeASLBD* genes.

LZL-region ($LX_6LX_3LX_6L$) was only found in *MeASLBD* proteins of Class I, and 20 *MeASLBD* protein sequences (*MeASLBD4*, *MeASLBD8*, *MeASLBD9*, *MeASLBD10*, *MeASLBD11*, *MeASLBD12*, *MeASLBD14*, *MeASLBD15*, *MeASLBD20*, *MeASLBD21*, *MeASLBD22*, *MeASLBD24*, *MeASLBD25*, *MeASLBD26*, *MeASLBD27*, *MeASLBD29*, *MeASLBD31*, *MeASLBD32*, *MeASLBD35*, *MeASLBD45*) contained a complete motif, while the remaining *MeASLBDs* contained an uncomplete motif. Moreover, most *MeASLBD* genes contained a complete ICG/GAS-region, except for *MeASLBD55*.

## Duplication, Collinearity, and synteny analysis

Genomic segmental and tandem duplications represent the two main driving forces of gene family expansion. MCScanX was applied to explore the gene duplication events in the *MeASLBD* gene family (Fig 4). Seven pairs of tandem duplicated genes were identified on chromosomes 1 (*MeASLBD31/27*), 5 (*MeASLBD26/35*), 7 (*MeASLBD13/55*), 10 (*MeASLBD24/6*), 12 (*MeASLBD20/14*), and 13 (*MeASLBD21/15*). 42 *MeASLBD* genes were segmentally duplicated, implying segmental duplication was a major driving force in the evolution of the *MeASLBD* gene family compared to tandem duplication. In addition, the $K_a/K_s$ for 34 *ASL/LBD* gene pairs were determined (S4 Table in S1 File). The $K_a/K_s$ ratio of *ASL/LBD* gene pairs varied from 0.0708 to 0.5342. The $K_a/K_s$ ratios were less than 1.0, indicating that these genes might have undergone purifying selection during the evolution process. Collinearity analysis revealed chromosomal duplication, translocation, and inversion in the *MeASLBD* gene family (Fig 5). Genome Synteny and Collinearity Analysis show that *ASL/LBD* is relatively conserved in *Arabidopsis* and cassava. *AtLBD* has a high degree of height with *MeASLBD*.

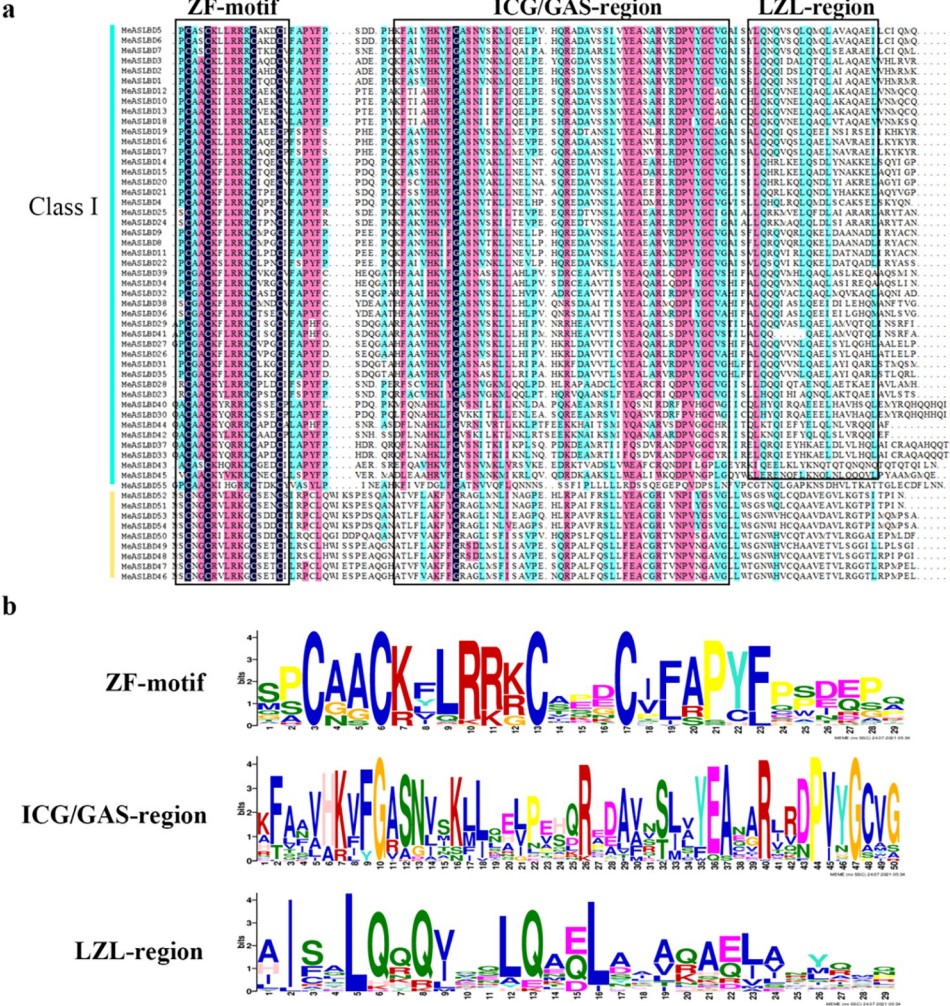

**Fig 3. Conserved domains of *MeASLBD* proteins.** (a) Multiple sequence alignment of *MeASLBD* conserved domains by DNAMAN. (b) Visualisation of the $CX_2CX_6CX_3C$ motif, ICG/GAS-region motif, and $LX_6LX_3LX_6L$ LZL-region.

## Analysis of putative cis-elements of the *MeASLBD* gene

Analysis of cis-regulatory elements in promoter regions helps understand the mode of gene regulation and predicts the functions of genes. The online tool Plant CARE was used to investigate the cis-elements of the *MeASLBD* genes within a 2000-bp upstream region (Fig 6). Among the seven known elements, the hormone-responsive, environmental stress-related, and development-related elements were the core physiological processes represented by the regulatory elements. The hormone-responsive elements present in the *MeASLBD* gene family were found to be related to auxin-producing genes (TGA-element, AuxRE, and AuxRR-core), gibberellin (GARE-motif, P-box, and TATC-box), salicylic acid (TCA-element), abscisic acid (ABRE), methyl jasmonic acid (CGTCA-motif, TGACG-motif), and ethylene (ERE), the elements related to ethylene, jasmonic methyl acid, and abscisic acid were abundant, indicating that these genes may respond to phytohormone signals and/or abiotic stresses. Environmental stress-related elements, including anaerobic induction, defence and stress responsiveness (TC-rich repeats), low-temperature responsiveness, and enhancer-like elements, are involved in

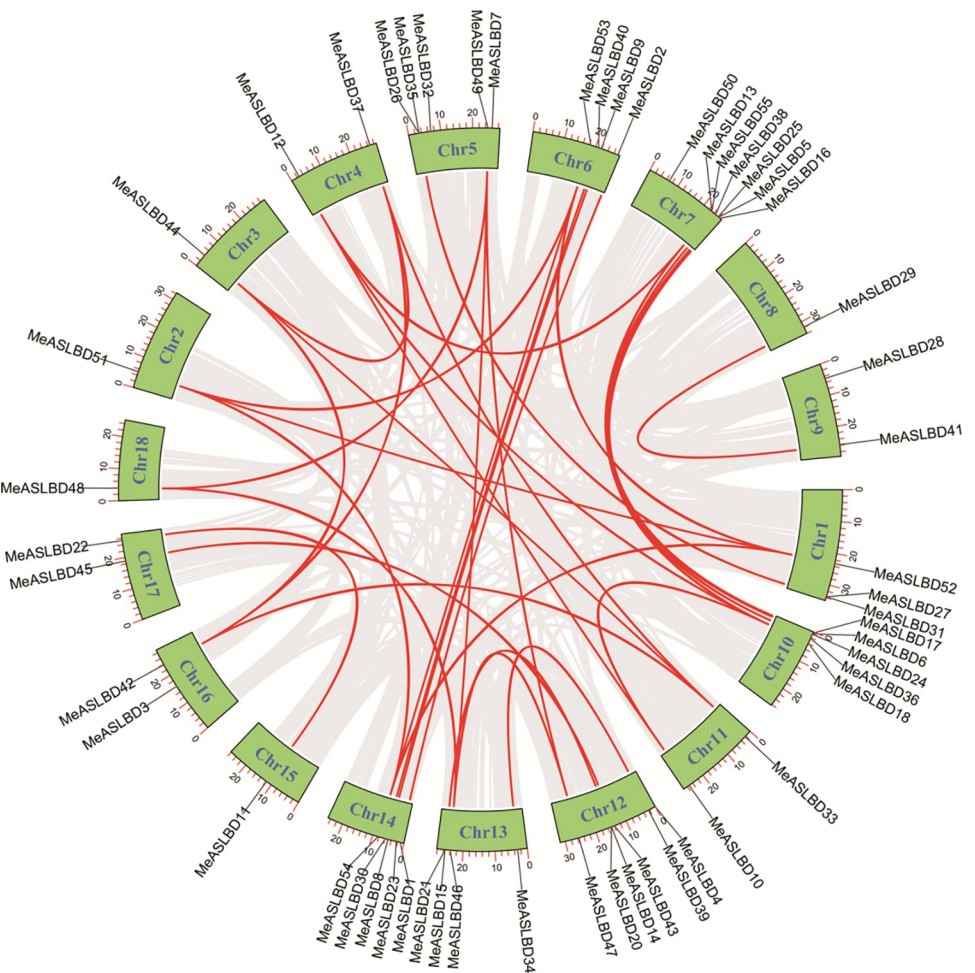

**Fig 4. Chromosomal distribution and interchromosomal relationships of *MeASLBD* genes.** The black lines show the position distribution of *MeASLBD* genes on 18 chromosomes. The red lines show the collinear gene pairs of *MeASLBDs*.

anoxic-specific inducibility (GC motif). In addition, meristem expression was found in development-related elements, suggesting that these genes might be involved in the development of plant meristems.

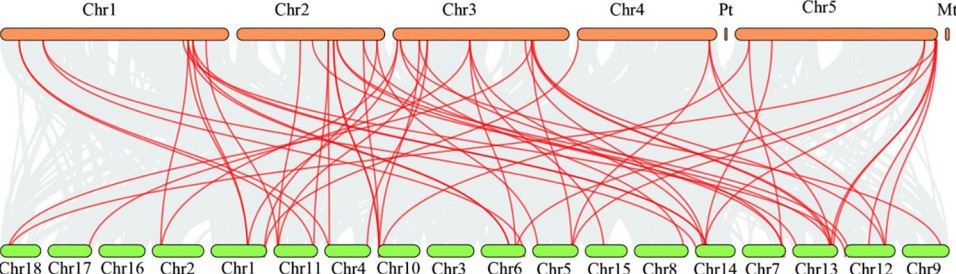

**Fig 5. Colinearity analysis.** Orange represents the chromosomes (Chr) of *Arabidopsis thaliana*, Green represents the chr of cassava, Red lines represent genes containing Homologous. Pt represents the Chloroplast, Mt represents mitochondrion. The collinearity analysis was conducted using MCScanX and visualized with TBtools.

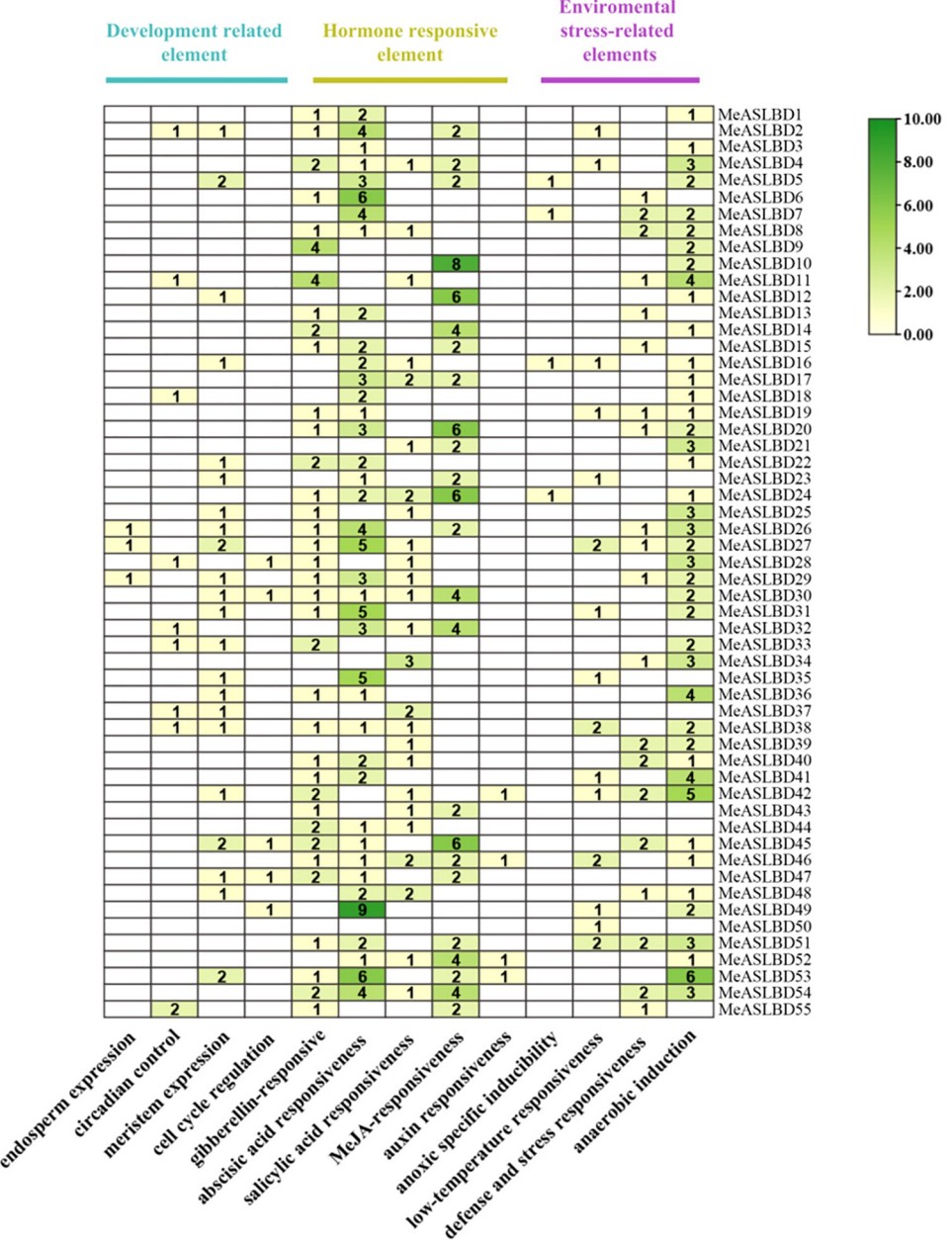

**Fig 6. The cis-elements of 2000 bp sequences upstream of *MeASLBD* gene analysed with the online tool PlantCARE (https://bioinformatics.psb.ugent.be/webtools/plantcare/html).** Tbtools software was used to search the cassava promoter elements, count the number of promoters with the same function, and used TBtools for visualization. The color bar represents the number of the putative cis-elements of the *MeASLBD* gene ranging from light green to dark green.

## Expression analysis of *MeASLBD* genes in different tissues and organs

The expression profiles of *MeASLBD* genes in different tissues (leaves and roots) of the cassava cultivars Arg7, W14, and KU50 were analysed ([Fig 7]). In cultivars Arg7, KU50, and W14; 1% (5/55), 9% (16/55), and 47% (26/55), respectively, of the *MeASLBD* genes were expressed in leaves and roots. *MeASLBD1, MeASLBD2, MeASLBD11, MeASLBD46, MeASLBD47* and

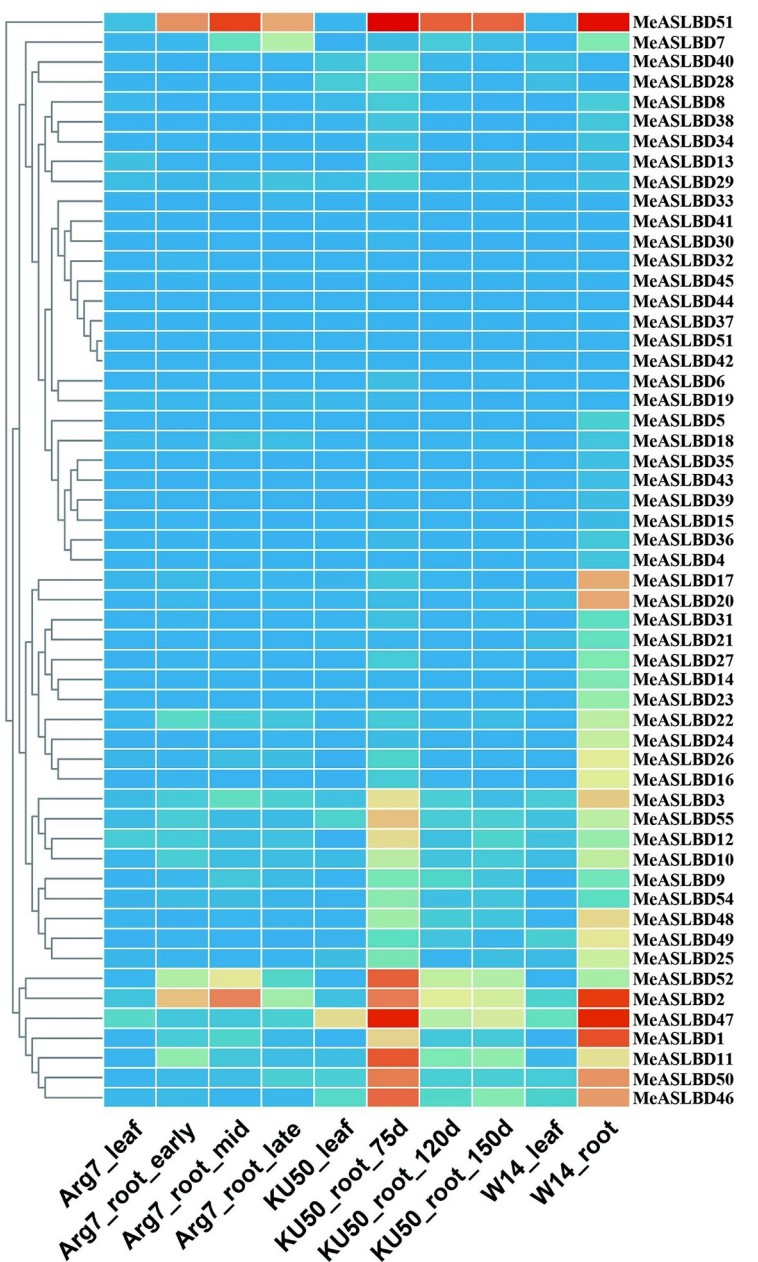

**Fig 7. Expression pattern analysis of *MeASLBD* transcription factors in roots and leaves.** Red indicates high expression, and blue indicates low expression. The colour scale shows RPKM values normalised by log2. Different tissues (leaves and roots) of the cassava cultivars Arg7, W14, and KU50 were analysed. The Arg7_leaf, KU50_leaf, and W14_leaf stand for leaves. And Arg7_root, KU50_root, and W14_root stand for root. Early, mid and late represent the stages. d = days.

*MeASLBD51* were highly expressed (RPKM value > 10) in roots and leaves, while 26 *MeASLBDs* had low or no expression in the three cassava cultivars.

Several genes displayed differential expression profiles between tissues or different cultivars. *MeASLBD1 and MeASLBD2* were highly expressed in the roots of W14 but at low levels in the leaves. *MeASLBD47* and *MeASLBD51* displayed higher expression in W14_root and KU50_root_75d. In addition, *MeASLBD47* was also expressed in KU50_leaf. These results suggest the genotype-dependent tissue expression of these genes.

## Expression profiles of the *MeASLBD* genes in response to drought and *Xam*CHN11 infection

The expression profiles of the 55 *MeASLBDs* in root and leaf tissues of two cassava cultivars (Arg7 and W14) under drought stress were investigated (Fig 8). The *MeASLBD* gene was expressed higher in roots than in leaves for both cultivars. Six *MeASLBDs* (*MeASLBD7*, *MeASLBD10*, *MeASLBD46*, *MeASLBD47*, *MeASLBD51* and *MeASLBD52*) had significant differential expression under drought stress. The *MeASLBD51* expression was significantly induced in Arg7, but decreased in W14 for the root tissues. Meanwhile, the expressions of *MeASLBD10*, *MeASLBD46*, and *MeASLBD47* were significantly altered in both roots and leaves. *MeASLBD7* and *MeASLBD52* were highly expressed in Arg7_CK12 root but were significantly downregulated under drought stress.

Besides the drought stress, the expressions of some *MeASLBDs* were also influenced under *Xam*CHN11 stress, including *MeASLBD10*, *MeASLBD12*, *MeASLBD46*, and *MeASLBD47*.

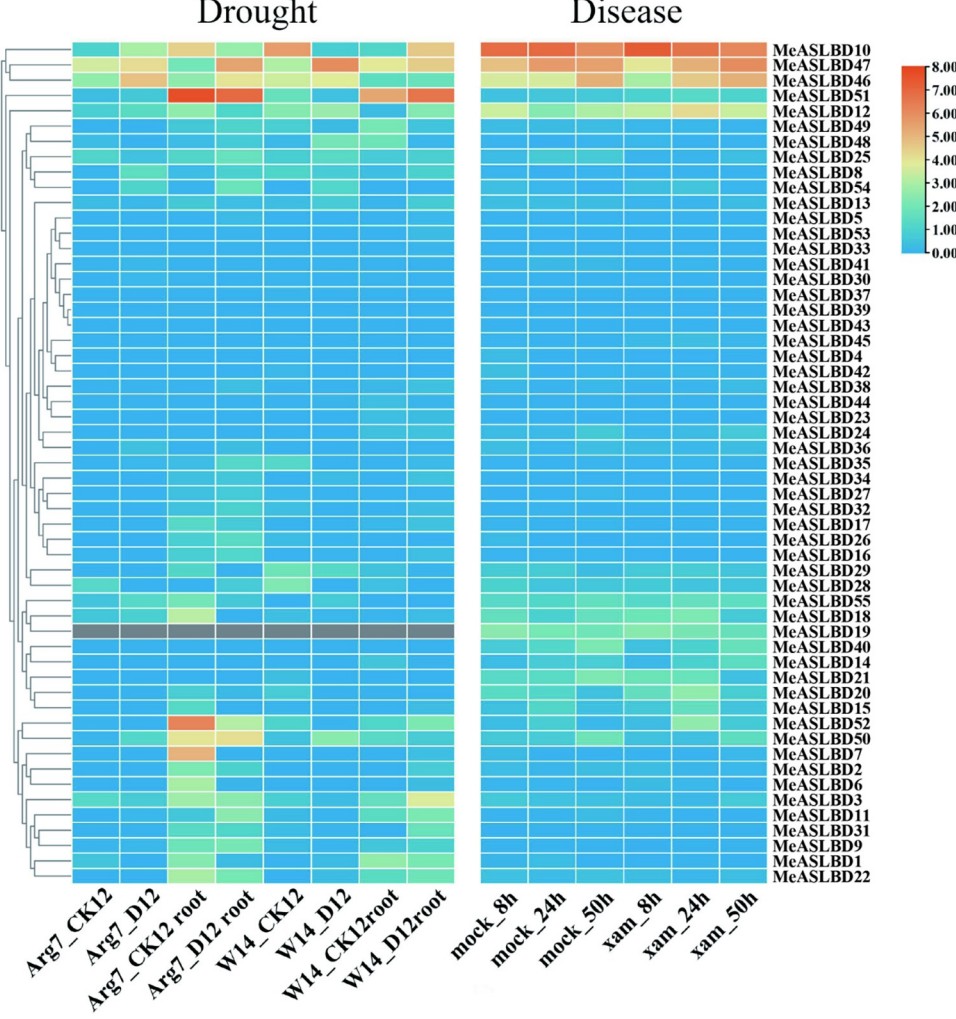

**Fig 8. Expression profiles of *MeASLBD* genes under drought and *Xam*CHN11 infection.** The ratio of the experimental group to the mock group was log2 transformed. W14_ck12l, W14D12, Arg7_ck12, and W14_CK12 represent leaves. And 14_ck12, W14D12 root, Arg7_ck12 root, and W14_CK12 root stands for root (S7 Table in S1 File). D stands for drought. (https://www.ncbi.nlm.nih.gov/biosample?Db=biosample&DbFrom=bioproject&Cmd=Link&LinkName=bioproject_biosample&LinkReadableName=BioSample&ordinalpos=1&IdsFromResult=246428).

Among these four genes, the *MeASLBD10* gene expression was steadily downregulated over infection time, whereas the *MeASLBD46* and *MeASLBD47* gene expressions were upregulated. Analyses of the transcriptome data of the cassava cultivars (Arg7 and W14) under stress treatments showed that some *ASL/LBD* genes respond significantly to abiotic and biotic stresses.

## Validation of six *MeASLBD* genes in response to *Xam*CHN11 *via* qRT-PCR

Six *MeASLBD* genes were randomly selected for qRT-PCR validation under a pathogenic bacterium (*Xam*) stress. The results of qRT-PCR showed that the *MeASLBD* genes (*MeASLBD1*, *MeASLBD2*, *MeASLBD12*, *MeASLBD13*, *MeASLBD46*, and *MeASLBD47*) had a different expression level at 1 h, 3 h, 6 h, and 9 h post-inoculation (Fig 9). The expression of *MeASLBD13* was upregulated at 6 h and 9 h post-inoculation but was downregulated at 1 h and 3 h post-inoculation. *MeASLBD46* was significantly upregulated at 1 h, 3 h, 6 h, and 9 h post-inoculation. *MeASLBD47* was upregulated progressively after inoculation but decreased slightly at 9 h post-inoculation. The expression levels of *MeASLBD1*, *MeASLBD2*, and *MeASLBD12* increased significantly at 6 h post-inoculation and then decreased at 9 h post-inoculation. These gene expression levels were consistent with the transcriptomic data. The results indicated that the most significant response of *MeASLBD* genes to pathogens occurred at 6 h post-inoculation. *MeASLBD*46, *MeASLBD13*, and *MeASLBD*47 expression levels were significantly expressed following treatment with *Xam*CHN11.

## Identification of candidate genes coexpressed with *MeASLBD* genes

One thousand four hundred forty genes contain a 5' `GCGGCG` 3' motif in the promoter of the cassava genome, a putative motif bound by *ASL/LBD* genes. GO enrichment analysis

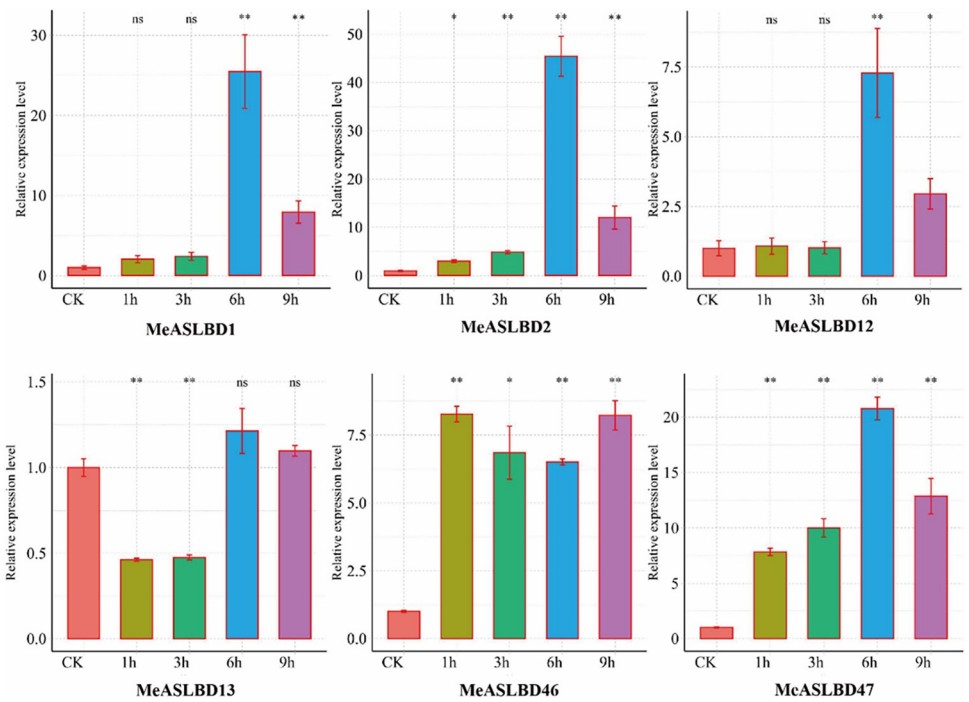

**Fig 9. Expression patterns of 6 *ASL/LBD* genes in different samples of *M. esculenta* (1h, 3h, 6h, 9h, and CK).** Accumulation levels of *MeASLBD* transcripts were determined by qRT-PCR (n = 3). EF1α was the internal control. Transcription levels were analysed using the comparative "$2^{-\Delta\Delta Ct}$" method. Symbols represent the mean ± SD. Groups designated by the same letter are not significantly different, while those with different letters ("*" or "***"). "**" ($p < 0.01$) and "*" ($0.01 < p < 0.05$).

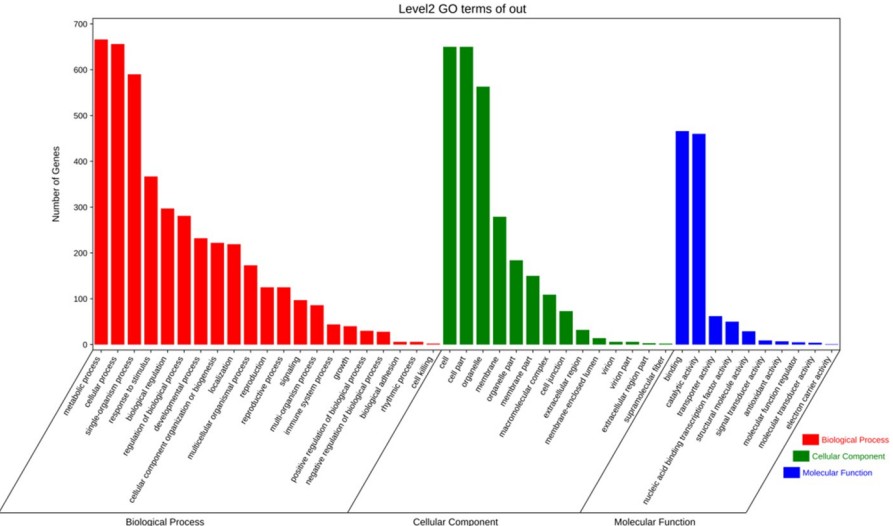

**Fig 10. GO enrichment analysis of *MeASLBDs* target genes.** Red is biological processes, green is cell composition, and blue is molecular functions.

revealed that these genes are mainly involved in metabolic, cellular, and biological processes. These genes are mainly allocated to cellular components for cells, cell parts, and organelles. Binding and catalytic activity are the main categories under molecular functions (Fig 10).

To further analyse the relationship between *MeASLBD* genes and the predicted downstream target genes, a set of 55 transcriptomes derived from *Xam*CHN11 infected cassava leaves was utilised to perform WGCNA co-expression analysis. The RPKM values of 55 *MeASLBD* genes and 1,450 target genes were determined. 347 genes with the expression coefficient of variation (CV) >1 and RPKM >1 were retained for WGCNA (S5 Table in S1 File). Highly interconnected gene sets known as modules were obtained *via* WGNCA analysis. The 347 genes could be divided into four modules *via* WGCNA, namely blue (67), brown (37), turquoise (217), and grey (25) (S6 Table in S1 File). Grey was considered irrelevant to the sample; therefore, this module was removed. The gene co-expression modules for the *MeASLBD* and target genes are shown in Fig 11. The enrichment analysis showed that 16 genes were involved in plant-pathogen interaction, 6 in the MAPK signalling pathway, and 14 in defence.

## Silencing of *MeASLBD47* reduced the lesion area and enabled differential expression of target genes

VIGS approach was applied to silence *MeASLBD47* gene. The silencing efficiency was examined using qRT-PCR. The results showed that the abundance of *MeASLBD47* in cassava species was reduced compared with NC (Negative Control), indicating that *MeASLBD47* was efficiently silenced (Fig 12A). Cassava was inoculated with pCsCMV-*AS2/LOB47*, pCsCMV-NC and 10 mM MgCl2 (Mock) for 3 days (Fig 12B). After three days, lesions appeared on the NC leaves, whereas pCsCMV-*AS2/LOB47* showed no obvious lesions. The result suggests that *MeASLBD47* silencing enhances plant resistance by reducing lesion areas. Through the co-expression network constructed using WGCNA, the possible downstream genes regulated by *MeASLBD* genes were predicted. The genes expressed in the regulatory network were detected based on VIGS (Fig 12C). Among them, the expressions of *HSFC1* and *GAD1* were significantly downregulated, whereas *HSP90* was significantly upregulated, indicating that they were affected by the effect of VIGS. After three days, *TFIIA-S* expression was

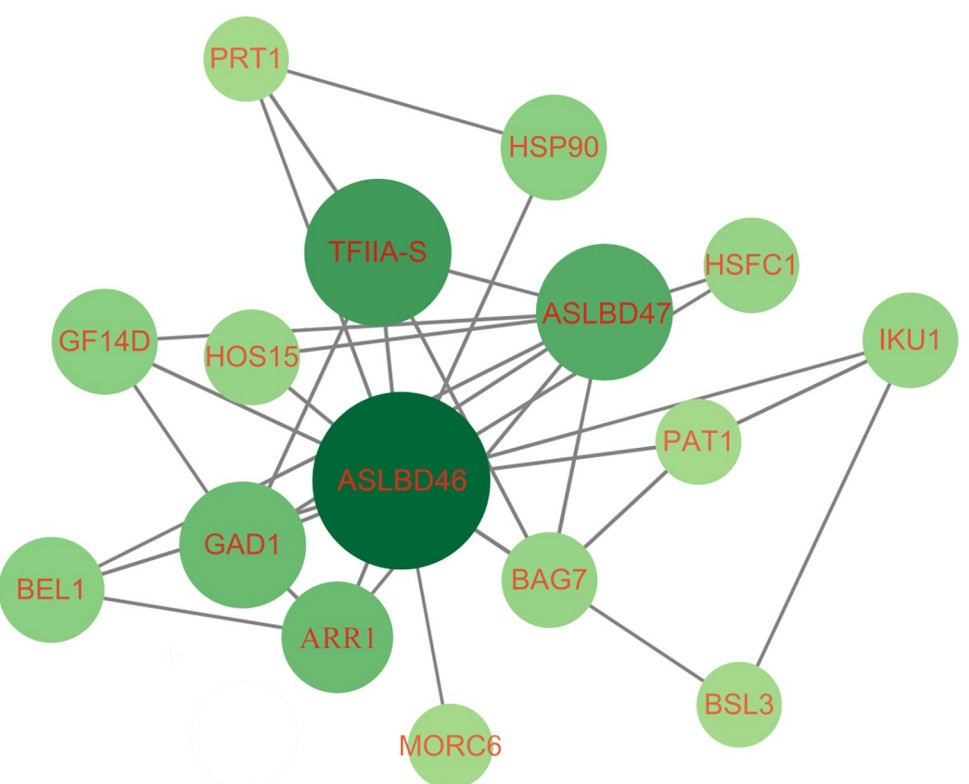

**Fig 11. The network of *MeASLBDs* and target genes in the turquoise module.** The genes in the WGCNA module showed expression patterns and consistent with those of *MeASLBDs*.

slightly affected. Overall, *MeASLBD47* might affect plant resistance by regulating *HSFC1* and *GAD1*.

## Discussion

Cassava (*M. esculenta. Crantz*) is a starch-rich, woody tuberous root crop that is important for food and as a potential bioenergy crop. Despite its importance in food and bioenergy, cassava cultivation is fraught with difficulties (abiotic and biotic stresses), resulting in decreased crop yield. Most are susceptible to pests and diseases threatening existing cultivars [63, 64]. *AS2/LOB* genes are widely present in plants and are important plant-specific process regulators [13]. To date, several *ASL/LBD* genes have been identified in *A. thaliana* (43) [9], *Eucalyptus grandis* (46) [65], *Glycine max* (90) [66], and other species. However, systematic studies of *MeASLBD* genes have not been reported so far. The availability of whole-genome sequences of *M. esculenta* helps in the genome-wide characterisation of *AS2/LOB* genes, which may be used to improve crop yield in the field. In this study, 55 *MeASLBD* genes were identified, indicating that the *AS2/LBD* gene family has largely retained a fixed function in the genetic evolution of different species. Two types of terminology were listed side by side (*ASLs* and *LBDs*). Such terminology might provide an advantage in discussing the evolutionary developmental biology of the *AS2/LOB* protein family [5]. The *AS2/LOB* gene family in *M. esculenta* is similar to the estimates for other reported plant species. The *MeASLBD* genes were divided into two classes (I and II) and nine subclasses (Ia~Ig, IIa, and IIb) based on the structure of the *LOB* domain (Fig 1). Forty-six *MeASLBD* genes (83%) belonged to Class I, and nine *MeASLBD* genes (17%) belonged to Class II. Previous studies also reported that 84% and 16% of the *ASL/LBD* genes in

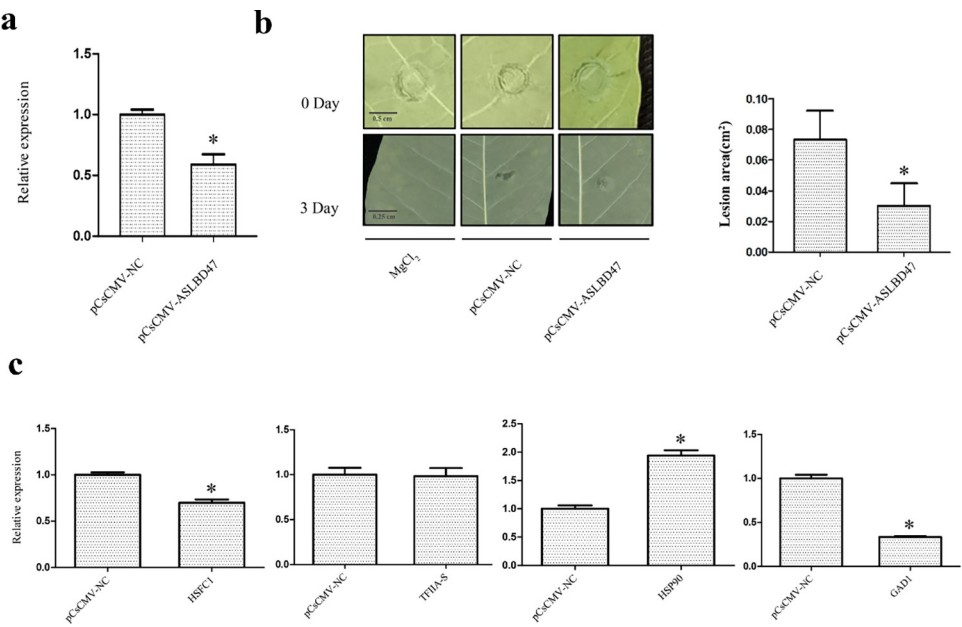

**Fig 12. Silencing of *MeASLBD47* in cassava plants using a cassava common mosaic virus (Cscmv)-based Virus-induced gene silencing (VIGS) system.** (a) Validation of the silencing efficiency of *MeASLBD* by qRT-PCR, pCsCMV-NC represents an empty vector. (b) The phenotypes of the cassava leaf infected with *Xam*CHN11 following inoculation with pCsCMV-*AS2/LOB*47 or empty vector (pCsCMV-NC), 10 mM MgCl2 (mock for *Xam*CHN11). The lesion area of cassava leaves was calculated using the ImageJ tool at 200 pixels/cm. (c) Differential expression of target genes in the WGCNA network upon silencing of *MeASLBD47*. "*": significantly different.

*A. thaliana* belonged to Classes I and II, respectively [11]. 98 *ASL/LBD* genes from *A. thaliana* and *M. esculenta* were further classified into nine subclasses (Ia-If and IIa-IIb) based on their phylogenetic relationships (Fig 1), and were consistent with those previously reported [67]. Homologous genes in the same class or subclass might share the same function. For instance, *ASL15/LBD17* was proposed to play a similar role to *ASL18/LBD16*, *ASL16/LBD29*, *ASL24/LBD33*, and *ASL20/LBD18* involved in auxin-induced lateral root formation, all of which belong to the close narrow clades of the tree [68–71]. These results imply that our finding of homologous genes may share similar functions.

The origin and evolutionary history of the *ASL/LBD* proteins remain largely unexplored. *ASL/LBD* proteins are streptophyte-specific proteins and likely evolved from the charophycean green algae [72]. Few TF families evolved within land plants, implying the origin of a new TF family is not critical for land plant evolution. Nevertheless, the increased diversity of several TF families, such as *ASL/LBD*, suggests that they may have been instrumental in terrestrialization [73]. Paleobotanical studies indicate that roots evolved at least twice independently during the evolution of land plants, once in lycophytes and once in euphyllophytes. Auxin promotes the initiation of postembryonic roots in both groups but from different cell types. *ASL/LBD* proteins act directly downstream of auxin in several euphyllophytes, including *Arabidopsis*, maize, and rice, and are conserved elements required for root initiation [72]. However, a clade with no lycophyte sequence specifically associated with root development was inferred, implying that root initiation in lycophytes and euphyllophytes is mediated by distinct genetic mechanisms downstream of auxin [17]. In contrast, leaves evolved independently in lycophytes and euphyllophytes [74]. The *MeALSBDs* evolutionary relationships provide an understanding and could contribute to determining the origin of the *ASL/LBD* protein family.

Structural analysis effectively extracts valuable information about gene duplication events and phylogenetic relationships within gene families. Gene duplication events, followed by neo-functionalisation, were enough to drive molecular diversification [73, 75]. *MeASLBD* genes have simple gene structures (Fig 2B). Most *MeASLBD* members within the same subgroup showed different exon/intron structure and gene length (Fig 2C), slightly different from *A. thaliana ASL/LBD* genes [9]. Therefore, *ASL/LBD* gene structures might not be relatively conserved in different species. In addition, cassava contained 55 *ASL/LBD* genes, 12 more than *Arabidopsis* and 35 fewer than soybean [60, 66]. Therefore, *ASL/LBD* genes vary from one species to another, consistent with previously reported *ASL/LBD* genes [9]. The $K_a/K_s$ ratio indicated that purifying selection had affected the cassava *AS2/LOB* domain (S4 Table in S1 File). In *Arabidopsis ASL/LBD* protein sequences comparison of the cloned cDNAs with those of the corresponding genes revealed that these genes were divided into five subtypes based on the positions of the introns in the coding regions (subtypes A to E). 40 *MeASLBD* genes contained splicing sites in their coding regions (Fig 2C). In comparison, 29 *Arabidopsis ASL/LBD* genes contained introns in their coding regions.

The *ASL/LBD* gene family is an important TF family in plant species. Therefore, the function of *MeASLBD* genes was investigated. Through promoter analysis, most of the *MeASLBD* promoter positions consisted of ABA, MeJA, and GA response cis-elements (Fig 6), which indicated that *MeASLBDs* might be involved in the plant phytohormone signal response. In *A. thaliana, AtASL/LBD20* is thought to play a role in the JA response [76], and *AtASL/LBD40* is downregulated under gibberellin treatment [77]. In cotton, *GhLBD123* was significantly upregulated under MeJA treatment [78]. These findings indicate that *MeASLBD* genes may be involved in hormone and stress responses.

The expression profiles of *MeASLBD* genes in different tissues (Fig 7) revealed that *MeASLBD* genes are mainly expressed in the roots, such as *MeASLBD2, MeASLBD47*, and *MeASLBD51*, indicating that they might be involved in cassava root development. The transcriptome expression profile of cassava under drought and disease stress was used to study the expression profiles of *MeASLBDs* (Fig 8). The results revealed that *MeASLBD46 and MeASLBD47* responded to drought stress and *Xam*CHN11 infection. *MeASLBD46* and *MeASLBD47* showed a strong response through qRT-PCR, indicating that they are associated with *Xam* infection response (Fig 9). The genes *MeASLBD46, MeASLBD47, AtASL/LBD37*, and *AtASL/LBD38* are homologous. *MeASLBD46* and *MeASLBD47* may thus be involved in nitrogen metabolism, anthocyanin synthesis [79], and stress response.

*AS2* can form complexes with other proteins to control different aspects of plant growth and development [1, 11, 80, 81]. *AS2* physically associates with *AS1* to form a repressor complex that regulates leaf polarity and morphology, inflorescence architecture and fertility, and shoot apical meristem differentiation [1, 11, 80, 81]. *AtASL2/LBD10* and *AtLBD28* are classified in class I in *Arabidopsis* [82]. In this study, *AtLBD28* was classified in class Id (Fig 1) alongside *MeASLBD28* and *MeALSBD23*. *AtLBD28* may influence leaf polarity and morphology, as well as shoot apical meristem differentiation [83]. Therefore, *AtLBD28, MeASLBD28*, and *MeALSBD23* may share similar functions. Moreover, *AtASL2/LBD10* participates in both microspore polarization before the first asymmetric division, and germ cell mitosis [84] *AtASL2/LBD10* mutants had a 12.7% aborted pollen grain yield, indicating that *AtASL2/LBD10* is important for *Arabidopsis* pollen development [85]. *AtASL2/LBD10* was classified in Class Ib (Fig 1), indicating that members such as *MeASLBD4* may share similar functions. In addition to their roles in plant development, some *Arabidopsis ASL/LBD* members play important roles in response to fungal pathogens (e.g., *Fusarium wilt*) and soil nematodes (e.g., *Meloidogyne incognita*) [76, 86]. For instance, *AtLBD20*, primarily expressed in the roots, is a negative regulator of *Fusarium wilt* resistance and a subset of jasmonate (JA) responses [76].

In this study, *AtLBD20*, *MeASLBD29*, and *MeASLBD41* were subgrouped into Class Ic (Fig 1), indicating that they may share the same function.

WGCNA is an effective tool for investigating the relationships between different gene sets (modules) [61]. Gene expression modules based on specific species have been efficiently used in diverse varieties. One thousand four hundred fifty genes were identified through promoter analysis. GO enrichment analysis (Fig 11) showed that these genes participate in different regulatory networks. Therefore, *AS2/LOB* proteins may regulate lateral root growth and plant signal transduction, metabolic processes, and stress responses. *MeASLBD46* and *MeASLBD47* responded to *Xam*CHN11 infection. In addition, fourteen genes were identified as target genes for *MeASLBD* regulation *via* the WGCNA co-expression network (Fig 11). In *Arabidopsis*, over 350 *AtLBD29* target genes have been identified to participate in the regulation of cell reprogramming during callus formation [13, 87, 88]. Some *MeASLBD* target genes are related to the plant pathogenic bacteria, such as *HSP90* [89], *TFIIA* [90], and *GAD1* [91].

VIGS tool was used to silence *MeASLBD47* gene and inoculate cassava with *Xam*CHN11 [42]. After 3 days of disease treatment, the water spots on leaves were reduced, and lesions developed slower than NC (Fig 12B). This indicates that the silencing of *MeASLBD47* enhances plant resistance against *Xam*CHN11. Simultaneously, qRT-PCR validation was performed on the co-expression network genes (Fig 12A). The result showed that *GAD1 and HSFC1* expression was inhibited and, on the contrary, enhanced the expression of *HSP90* (Fig 12C). *MeASLBD47* might regulate downstream target genes *GAD1*, *HSFC1*, and *HSP90* to enhance plant resistance. In addition, transcriptome analysis of *AS2/LOB37* mutant material in *Arabidopsis* [79] found that *LRR* (*At1g66090*) was significantly upregulated when *AS2/LOB37* was deleted. Therefore, the *LRR* gene in cassava was detected, and the results were consistent with *Arabidopsis* (S2 Fig in S1 File). These findings provide useful clues for further investigations into gene regulatory networks involved in cassava disease resistance.

## Conclusions

Fifty-five *ASL/LBD* genes were identified from *M. esculenta* and distributed unevenly among the chromosomes. The *MeASLBD* gene family is classified into two categories based on gene structure and phylogeny: Class I (46) and Class II (9). Collinearity analysis and the Ks/Ks values indicated that purifying selection was the main force driving the evolution of the *MeASLBD* gene family. According to the transcriptome data of cassava under biological and abiotic stress, *MeASLBD46* and *MeASLBD47* were found to have a strong disease and drought response. To better understand the regulatory function of *MeASLBDs*, the target genes of *MeASLBDs* were screened and revealed the disease-related genes of *HSP90*, *TFIIA-S*, and *GAD1*. The latter result indicates that *MeASLBD46* and *MeASLBD47* may participate in the plant response to stress by regulating these target genes. Furthermore, these findings provide valuable information for subsequent elucidation of the role of the *ASL/LBD* genes.

## Supporting information

**S1 File. Contains all the supporting tables and figures.**
(DOCX)

## Acknowledgments

The authors thank AJE (https://www.aje.cn) for providing linguistic assistance during the preparation of this manuscript.

## Author Contributions

**Conceptualization:** Yiming Mao, Yinhua Chen.

**Data curation:** Xiaowen Yao, Yijie Zhang, Yiwei Ye.

**Formal analysis:** Assane Hamidou Abdoulaye, Jiming Song, Xiaowen Yao, Kai Luo.

**Investigation:** Xiaowen Yao, Kai Luo.

**Methodology:** Yu Gao, Kai Luo.

**Software:** Wei Xia.

**Supervision:** Yinhua Chen.

**Validation:** Kai Luo.

**Writing – original draft:** Yiming Mao, Assane Hamidou Abdoulaye, Wei Xia, Yinhua Chen.

**Writing – review & editing:** Yiming Mao, Assane Hamidou Abdoulaye, Yinhua Chen.

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
