## [Decision Letter · Decision Letter 0]

27 Apr 2022

PONE-D-22-03287Genome-wide characterisation of cassava LATERAL ORGAN BOUNDARIES domain genes revealed LBD47 involved in bacterial blight defensePLOS ONE

Dear Dr. chen,

Thank you for submitting your manuscript to PLOS ONE. After careful consideration, we feel that it has merit but does not fully meet PLOS ONE’s publication criteria as it currently stands. Therefore, we invite you to submit a revised version of the manuscript that addresses the points raised during the review process.

We look forward to receiving your revised manuscript.

Kind regards,

Farrukh Azeem

Academic Editor

PLOS ONE

Journal Requirements:

“This study was supported by the Natural Science Foundation of Hainan Province (320QN193) and the China Agriculture Research System (CARS-11).”

“This study was financially supported by National Key Research and Development Program (2018YFD1000500) and China Agriculture Research System (CARS-11-hncyh).”

 “This study was supported by the Natural Science Foundation of Hainan Province (320QN193) and the China Agriculture Research System (CARS-11).”

Reviewers' comments:

Reviewer's Responses to Questions

**Comments to the Author**

1. Is the manuscript technically sound, and do the data support the conclusions?

Reviewer #1: Partly

Reviewer #2: Partly

2. Has the statistical analysis been performed appropriately and rigorously? 

Reviewer #1: No

Reviewer #2: No

3. Have the authors made all data underlying the findings in their manuscript fully available?

Reviewer #1: Yes

Reviewer #2: Yes

4. Is the manuscript presented in an intelligible fashion and written in standard English?

Reviewer #1: No

Reviewer #2: No

5. Review Comments to the Author

Reviewer #1: The authors reported the AS2/LOB gene family in cassava (Manihot esculenta), in which members contain plant specific AS2/LOB domain. They identified and characterized 55 ASL/LBD genes (MeLBD1 to MeLBD55) from cassava genome using hidden Markov model profiles. They showed that the expression profiles of these genes were highly diverse. Weighted gene co-expression network analysis (WGCNA) of target genes and promoter analysis suggest that some members of cassava MeLBDs may be involved in hormone and stress responses. Furthermore, they described that the promoter analysis suggested that cassava MeLBDs may be involved in the plant phytohormone signal response.

Major points:

(1) Genome-wide characterisation of the AS2/LOB gene family should be very useful for understanding the function of these genes in cassava. To refine these analysis, they should refer to four papers as following on this subject.

Bowman et al., 2017. Insights into Land Plant Evolution Garnered from the Marchantia polymorpha Genome. Cell 171, 287-304 doi.org/10.1016/j.cell.2017.09.030

Machida Y., Suzuki T., Sasabe M., Iwakawa H., Kojima S., Machida C. Arabidopsis ASYMMETRIC LEAVES2 (AS2): roles in plant morphogenesis, cell division, and pathogenesis J. Plant Research 2022 135:3–14 (doi.org/10.1007/s10265-021-01349-6）

Coudert, Y.; Dievart, A.; Droc, G.; Gantet, P. ASL/LBD phylogeny suggests that genetic mechanisms of root initiation downstream of auxin are distinct in lycophytes and euphyllophytes. Mol. Biol. Evol. 2013, 30, 569–572.

Matsumura Y, Iwakawa H, Machida Y, Machida C (2009) Characterization of genes in the ASYMMETRIC LEAVES2/LATERAL ORGAN BOUNDARIES (AS2/LOB) family in Arabidopsis thaliana, and functional and molecular comparisons between AS2 and other family members. Plant J 58:525–537

According to these papers, LATERAL ORGAN BOUNDARIES domain genes should be written ASYMMETRIC LEAVES2-like/LATERAL ORGAN BOUNDARIES DOMAIN (ASL/LBD) genes, MeLBD should be MeASLBD and the LBD gene family should be the AS2/LOB gene family.

In Phylogenetic trees of figure 1, all AtLBD should be written AtASL/LBD, for example, AtASL1/LBD36, AtASL2/LBD10, AtASL3/LBD25, AtASL4/LOB, AtAS2/LBD6, which are classified into Class Ib in this paper. If so, it should be very useful to understand the evolutionary relationship between the ASL/LBD proteins of cassava and those of arabidopsis. In addition, it should be useful for the functionary relationship between them, because the authors can recognize that those with similar numbers are functionally close to each other in arabidopsis.

In figure 1, MeLBD should be MeASLBD, for example, MtASLBD1, MtASLBD2, MtASLBD3 like Marchantia (Bowman et al., 2017).

(2) AS shown in figure 2, it is very interesting that members of the MeASLBD gene family have one to four exons. Class Ia contain one putative splicing site in the AS2/LOB domain at the same site as that in arabidopsis (see Figure 1b in Matsumura et al., 2009) . In arabidopsis a comparison of the sequences of the cloned cDNAs with those of the corresponding genes revealed that 29 genes contained introns in their coding regions, and these genes could be divided into five subtypes on the basis of the positions of the introns in the coding regions (subtypes A–E). The subtype in arabidopsis seems to be corresponded to subtypes in cassaba. They should describe the relationship of exons/introns in their AS2/LOB domain regions between cassaba and arabidopsis.

(3) As shown in Figure 3, multiple sequence alignment of the MeAS2/LOB domain sequences using DNAMAN showed that all sequences contained three subdomeins named C Block, GAS block and L-rich Block. According to the recent two papers as follows, the authors should change the name of subdomain in the AS2/LOB domain. C Block should be written ZF-motif, GAS block should be ICG/GAS-region, L-rich Block should be LZL-region

Machida et al., (2022) J. Plant Research 135:3–14 (doi.org/10.1007/s10265-021-01349-6）

Chen WF, Wei XB, Rety S, Huang LY, Liu NN, Dou SX, Xi XG (2019) Structural analysis reveals a “molecular calipers” mechanism for a LATERAL ORGAN BOUNDARIES DOMAIN transcription factor protein from wheat. J Biol Chem 294:142–156

(4) The authors described that potential target genes were identified via promoter analysis, and a co-expression network was constructed along with MeLBDs based on a transcriptome dataset derived from pathogen treatment. They found that the MeLBD47 gene expression was highly induced following inoculation with cassava bacterial blight pathogen (XamCHN11), selected for functional analysis. And then the VIGS technique was applied to determine the function of the MeLBD47 gene following XamCHN11 treatment.

They described that one thousand four hundred forty genes contain a 5’ GCGGCG 3’ motif in the only promoter of the cassava genome, a putative motif bound by ASL/LBD genes. Although some members of the AS2/LOB family have been shown interaction to 5’ GCGGCG 3’ motif (Husbands et al., 2007; Vial-Pradel et al., 2018; Chen et al., 2018), function of interaction to 5’ GCGGCG 3’ motif is not known yet. Vial-Pradel et al. found that AS2 binds specifically the sequence containing 5’ GCGGCG 3’ motif in exon 1 of ETT/ARF3 (direct target of AS2), and that the binding requires the ZF-motif in AS2 that is structurally similar to the zinc finger-CxxC domain in vertebrate DNA methyltransferase1. In addition, Vial-Pradel et al. showed that AS2 did not bind to DNA sequence containing 5’ GCGGCG 3’ like motif of promoter region of ETT/ARF3.

The authors may examine binding to DNA sequence containing 5’ GCGGCG 3’ motif in candidate of target of MeASLBD47 and/or discuss about function of binding to DNA sequence containing 5’ GCGGCG 3’ motif.

Vial-Pradel S., Keta S., Nomoto M., Luo L., Takahashi H., Suzuki M., Yokoyama Y., Sasabe M., Kojima S., Tada Y., Machida Y., Machida C. Arabidopsis zinc-finger-like protein ASYMMETRIC LEAVES2 (AS2) and two nucleolar proteins maintain gene body DNA methylation in the leaf polarity gene ETTIN (ARF3). Plant Cell Physiol. 59(7):1385-1397 (2018).

(5) In Figure 11b, the authors showed that the leaves of pCsCMV-LBD47 plants were inoculated with XamCHN11, and pCsCMV-LBD47 showed no obvious lesions after three days, although lesions appeared in the pCsCMV-NC (control). They should show quantitative data.

(6) Line 349, The authors wrote “Among them, the expressions of HSFC1 HSP90 and GAD1 were significantly downregulated, indicating that they were affected by the effect of VIGS.” However, HSP90 was not downregulated, but upregulated.

(7) There is no Figure legends. The authors should write Figure legends for Fig 1-11.

(8) In References, The authors did not write name of journal. They should write name of Journal of each reference.

(9) Arabidopsis Thaliana should be Arabidopsis thaliana.

Reviewer #2: This paper “Genome-wide characterisation of cassava LATERAL ORGAN BOUNDARIES domain genes revealed LBD47 involved in bacterial blight defense” investigates the LBD genes (MeLBD1 to MeLBD55) in cassava genome using hidden Markov model profiles. A lot of papers have been published on cassava plants infected with Xanthomonas axonopodis. The work showed methodological lacks where necessary, unnecessary statements, and weak discussion. The experimental design data has been poorly explored without any statistical analysis and result description. Moreover, the manuscript has a lot of language and editing issues, which should be corrected and needs extensive revision. In some lines, the inappropriate use of English language makes the manuscript really hard to follow.

Title:

The inappropriate use of English language makes difficult to follow the title. It should be precise and understandable.

Abstract:

Abstract is not well written, there is no proper background, methods, directly results are discussed without any conclusion. There are lots of English language issues that makes the abstract really hard to follow. Some very basic scientific writing errors are there, such as on line 23 “Arabidopsis Thaliana” is written. It should be written as Arabidopsis thaliana.

Keywords:

Authors should use full name instead of abbreviation in this part

Introduction:

The Introduction is poorly written and has many language issues, which need to be corrected. It is advised to get help of English editing service. The literature discussed in one sentence is mostly disconnected from the next sentence, hampering the reading fluency e.g. from Line 43 to 52, 55 to 57, 66 to 67, etc. At some places the meaning of sentence is ambiguous. The information provided need improvement by adding literature relevant to the study conducted. The study objectives are lacking. Authors must clearly present the hypothesis of their study.

Methods:

The methods description also needs improvements. The authors should use scientific language to write protocols and procedures. More clarity is needed in this section like:

• How the Xanthomonas axonopodis infection was given and at which CFU?

• How symptoms were monitored post-inoculation and evaluated according to a severity scale?

• How the HMM profile was created?

• How authors analyzed the conserved domains?

• Which model authors used to construct the phylogenetic tree?

• What was the optimum width of each motif?

• From where authors retrieve the files for analyzing exon-intron structure?

• Authors should express the syntenic relationship of the gene pairs and their respective loci in cassava genome.

• From where authors retrieve GFF3 file for chromosomal location?

• Microcollinearity is valuable to understand the investigation of gene loss during evolution or the evolution of specific genes in a local region. Therefore, authors must do TGT (Triticeae-Gene Tribe) analysis to trace the origin history of the target gene, and also analyze gene pair.

Results:

Result part need substantial improved. The inappropriate use of English language makes the results hard to follow. For example, from lines 178 to 180, etc. Results are ambiguous. The authors must double check the information given in the manuscript as the methodology section saying something different than what is explained in results section including figures.

• The information given in Table 1 is already available on “iTAK - Plant Transcription factor & Protein Kinase Identifier and Classifier”. What new authors have done?

• Figures are not clear. Authors must add high resolution figures to make them understandable. Authors must double check whether the information in the figures are same as they have been written in the results section.

• There are no figure legends. Authors are advised to add them.

• Were the LBD genes groups evenly distributed among chromosomes or different homologous groups?

• Methodology and results of synteny analysis are missing while there is heading in the results.

• Authors must analyze the duplication event

• The results of Exon–intron structure divergences are missing. Authors have only discussed exons but there is no information regarding the introns in all these genes.

• The authors must correlate the phylogenetic relationship among LBDs with exon–intron structure

• In methodology, authors wrote that “The cassava cultivar used in this study was South China 8 (SC8)” while in the results the authors wrote “The expression profiles of MeLBDs in different tissues (leaves and roots) of the cassava cultivar Argentina 7 (Arg7), wild subspecies (W14) and variety Kasetsart University 50 (KU50) were analysed”. Which information is correct?

• What does the heatmap in Figure 6 shows? There is no methodology for this. Results are different from what is said in the methodology.

• The results described in “Expression analysis of MeLBD genes in different tissues and organs” are difficult to understand as the figure which is cited in this don’t show the similar information.

• There is no methodology for drought. How it appear in the results section?

• The transcriptome analysis was done with which treatments and on which plants?

• When silencing of MeLBD47 gene was done? What was its methodology?

• How the statistical analysis was performed?

• What does the stars on the figure 8 show?

Discussion:

Discussion is very poorly written. This section needs substantial improvement in terms of English language and relevant review of literature. In its current form, it is written much more like introduction and results. In text citation also need revision.

Overall, the manuscript seems to be the first rough draft which has to be tidied up in many places and this manuscript in its current form cannot be considered for publication. The manuscript needs extensive revisions before considering it for publication.

6. PLOS authors have the option to publish the peer review history of their article (what does this mean?). If published, this will include your full peer review and any attached files.

Reviewer #1: No

Reviewer #2: **Yes: **Hamid Manzoor

---

## [Author Response · Author response to Decision Letter 0]

15 Jul 2022

Journal Requirements:

Response: Thanks for providing us with the PLOS ONE style templates. The manuscript has been thoroughly improved following PLOS ONE's style requirements.

"This study was supported by the Natural Science Foundation of Hainan Province (320QN193) and the China Agriculture Research System (CARS-11)."

Response: This study was supported by National Key Research and Development Program (2018YFD1000500), the Natural Science Foundation of Hainan Province (320QN193) and China Agriculture Research System (CARS-11-hncyh).

National Key Research and Development Program (2018YFD1000500) Funding had role in study design and material collection; the Natural Science Foundation of Hainan Province (320QN193) had role in the research of qPCR data collection and analysis; China Agriculture Research System (CARS-11-hncyh) founders had role in RNA-Seq data collection and analysis, manuscript preparation. 

"This study was financially supported by National Key Research and Development Program (2018YFD1000500) and China Agriculture Research System (CARS-11-hncyh)."

 "This study was supported by the Natural Science Foundation of Hainan Province (320QN193) and the China Agriculture Research System (CARS-11)."

Response: Thanks for pointing it out. We have removed the funding from the manuscript.

Upon re-submitting your revised manuscript, please upload your study's minimal underlying data set as either Supporting Information files or to a stable, public repository and include the relevant URLs, DOIs, or accession numbers within your revised cover letter. For a list of acceptable repositories, please see http://journals.plos.org/plosone/s/data-availability#loc-recommended-repositories. Any potentially identifying patient information must be fully anonymized.

Response: The data used for the analysis in this study are available within the article and supplementary materials. Please see supplementary material.

5. PLOS requires an ORCID iD for the corresponding author in Editorial Manager on papers submitted after December 6th, 2016. Please ensure that you have an ORCID iD and that it is validated in Editorial Manager. To do this, go to 'Update my Information' (in the upper left-hand corner of the main menu), and click on the Fetch/Validate link next to the ORCID field. This will take you to the ORCID site and allow you to create a new iD or authenticate a pre-existing iD in Editorial Manager. Please see the following video for instructions on linking an ORCID iD to your Editorial Manager account: https://www.youtube.com/watch?v=_xcclfuvtxQ

Response: My ORCID iD 0000-0002-9232-7590 has now been linked to my PlosOne profile.

Response: Not applicable

7. We notice that your manuscript file was uploaded on Jan 30 2022. Please can you upload the latest version of your revised manuscript as the main article file, ensuring that does not contain any tracked changes or highlighting. This will be used in the production process if your manuscript is accepted. Please follow this link for more information: http://blogs.PLOS.org/everyone/2011/05/10/how-to-submit-your-revised-manuscript/

Response: We had uploaded the latest version of the clean manuscript as the main article file.

8. Please amend the title either on the online submission form or in your manuscript so that they are identical. 

Response: Sorry for the mistake. We had revised the online submission form. The tile of the form and the manuscript were identical.

---

## [Decision Letter · Decision Letter 1]

10 Aug 2022

PONE-D-22-03287R1A genome-wide analysis of the ASYMMETRIC LEAVES2-LIKE/LATERAL ORGAN BOUNDARIES (ASL/LBD) gene family in cassava (Manihot esculenta) and expression profile under biotic and abiotic stressPLOS ONE

Dear Dr. chen,

Thank you for submitting your manuscript to PLOS ONE. After careful consideration, we feel that it has merit but does not fully meet PLOS ONE’s publication criteria as it currently stands. Therefore, we invite you to submit a revised version of the manuscript that addresses the points raised during the review process. Please submit your revised manuscript by Sep 24 2022 11:59PM. If you will need more time than this to complete your revisions, please reply to this message or contact the journal office at plosone@plos.org. Please include the following items when submitting your revised manuscript:A rebuttal letter that responds to each point raised by the academic editor and reviewer(s). You should upload this letter as a separate file labeled 'Response to Reviewers'.A marked-up copy of your manuscript that highlights changes made to the original version. You should upload this as a separate file labeled 'Revised Manuscript with Track Changes'.An unmarked version of your revised paper without tracked changes. You should upload this as a separate file labeled 'Manuscript'.

We look forward to receiving your revised manuscript.

Kind regards,

Farrukh Azeem

Academic Editor

PLOS ONE

Reviewers' comments:

Reviewer's Responses to Questions

**Comments to the Author**

1. If the authors have adequately addressed your comments raised in a previous round of review and you feel that this manuscript is now acceptable for publication, you may indicate that here to bypass the “Comments to the Author” section, enter your conflict of interest statement in the “Confidential to Editor” section, and submit your "Accept" recommendation.

Reviewer #1: (No Response)

Reviewer #2: All comments have been addressed

2. Is the manuscript technically sound, and do the data support the conclusions?

Reviewer #1: Partly

Reviewer #2: Yes

3. Has the statistical analysis been performed appropriately and rigorously? 

Reviewer #1: I Don't Know

Reviewer #2: Yes

4. Have the authors made all data underlying the findings in their manuscript fully available?

Reviewer #1: Yes

Reviewer #2: Yes

5. Is the manuscript presented in an intelligible fashion and written in standard English?

Reviewer #1: No

Reviewer #2: Yes

6. Review Comments to the Author

Reviewer #1: The authors have corrected some of the things I pointed out. However, there are still many shortcomings. In addition, there are many inadequate understandings of important papers published in the past.

<comment>

Although I suggested four papers to be concerned, the authors did not cite the three papers as follows. They should read precisely, cite them and add them in References.

Bowman et al., 2017. Insights into Land Plant Evolution Garnered from the Marchantia polymorpha Genome. Cell 171, 287-304 doi.org/10.1016/j.cell.2017.09.030

Machida Y., Suzuki T., Sasabe M., Iwakawa H., Kojima S., Machida C. Arabidopsis ASYMMETRIC LEAVES2 (AS2): roles in plant morphogenesis, cell division, and pathogenesis. J. Plant Research 2022 135:3–14 (doi.org/10.1007/s10265-021-01349-6)

Coudert, Y.; Dievart, A.; Droc, G.; Gantet, P. ASL/LBD phylogeny suggests that genetic mechanisms of root initiation downstream of auxin are distinct in lycophytes and euphyllophytes. Mol. Biol. Evol. 2013, 30, 569–572.

<comment>

Although I suggested that all AtLBD should be written AtASL/LBD in Figure 1, the authors wrote only 15 genes out of 43 genes as AtASL/LBD. They should change all AtLBD to AtASL/LBD.

<comment>

Although I pointed, the authors did not change them in Figure 3. They should replace. C Block should be written ZF-motif, GAS block should be ICG/GAS-region, L-rich Block should be LZL-region.

<comment>

Although the authors described that they could not achieve this analysis due to the timing given, they should understand precisely and cite the following two papers at least.

Vial-Pradel S., Keta S., Nomoto M., Luo L., Takahashi H., Suzuki M., Yokoyama Y., Sasabe M., Kojima S., Tada Y., Machida Y., Machida C. Arabidopsis zinc-finger-like protein ASYMMETRIC LEAVES2 (AS2) and two nucleolar proteins maintain gene body DNA methylation in the leaf polarity gene ETTIN (ARF3). Plant Cell Physiol. 59(7):1385-1397 (2018).

Chen WF, Wei XB, Rety S, Huang LY, Liu NN, Dou SX, Xi XG (2019) Structural analysis reveals a "molecular calipers" mechanism for a LATERAL ORGAN BOUNDARIES DOMAIN transcription factor protein from wheat. J Biol Chem 294:142–156

<comment>

They did not complete to correct yet. They should correct all.</comment></comment></comment></comment></comment>

Reviewer #2: I have gone through the revised version of manuscript entitled "A genome-wide analysis of the ASYMMETRIC LEAVES2-LIKE/LATERAL ORGAN BOUNDARIES (ASL/LBD) gene family in cassava (Manihot esculenta) and expression profile under biotic and abiotic stress" (PONE-D-22-03287R1). The authors have made satisfactory changes in the revised version of the manuscript. The manuscript in its current form can be ACCEPTED for publication in PlosONE.

7. PLOS authors have the option to publish the peer review history of their article (what does this mean?). If published, this will include your full peer review and any attached files.

Reviewer #1: No

Reviewer #2: **Yes: **Dr. Hamid Manzoor

---

## [Author Response · Author response to Decision Letter 1]

19 Sep 2022

Reviewer #1: The authors have corrected some of the things I pointed out. However, there are still many shortcomings. In addition, there are many inadequate understandings of important papers published in the past.

Although I suggested four papers to be concerned, the authors did not cite the three papers as follows. They should read precisely, cite them and add them in References.

Bowman et al., 2017. Insights into Land Plant Evolution Garnered from the Marchantia polymorpha Genome. Cell 171, 287-304 doi.org/10.1016/j.cell.2017.09.030

Machida Y., Suzuki T., Sasabe M., Iwakawa H., Kojima S., Machida C. Arabidopsis ASYMMETRIC LEAVES2 (AS2): roles in plant morphogenesis, cell division, and pathogenesis. J. Plant Research 2022 135:3–14 (doi.org/10.1007/s10265-021-01349-6)

Coudert, Y.; Dievart, A.; Droc, G.; Gantet, P. ASL/LBD phylogeny suggests that genetic mechanisms of root initiation downstream of auxin are distinct in lycophytes and euphyllophytes. Mol. Biol. Evol. 2013, 30, 569–572.

Response: Thanks for pointing out these omissions. Indeed, these references outstandingly studied the origin and evolution of the LOB domain. Therefore, these references were cited throughout the revised manuscript. 

Although I suggested that all AtLBD should be written AtASL/LBD in Figure 1, the authors wrote only 15 genes out of 43 genes as AtASL/LBD. They should change all AtLBD to AtASL/LBD.

Response: Thanks for pointing it out. All AtLBD were changed to AtASL/LBD. Please see figure 1

Although I pointed, the authors did not change them in Figure 3. They should replace. C Block should be written ZF-motif, GAS block should be ICG/GAS-region, L-rich Block should be LZL-region.

Response: We apologize for this omission. C Block, GAS block, and L-rich Block were changed into ZF-motif, ICG/GAS-region, and LZL-region, respectively, throughout the manuscript and in figure 3. Please see figure 3.

Although the authors described that they could not achieve this analysis due to the timing given, they should understand precisely and cite the following two papers at least. 

Vial-Pradel S., Keta S., Nomoto M., Luo L., Takahashi H., Suzuki M., Yokoyama Y., Sasabe M., Kojima S., Tada Y., Machida Y., Machida C. Arabidopsis zinc-finger-like protein ASYMMETRIC LEAVES2 (AS2) and two nucleolar proteins maintain gene body DNA methylation in the leaf polarity gene ETTIN (ARF3). Plant Cell Physiol. 59(7):1385-1397 (2018).

Chen WF, Wei XB, Rety S, Huang LY, Liu NN, Dou SX, Xi XG (2019) Structural analysis reveals a "molecular calipers" mechanism for a LATERAL ORGAN BOUNDARIES DOMAIN transcription factor protein from wheat. J Biol Chem 294:142–156 

Response: Thanks for these suggestions. These references were cited throughout the revised manuscript. 

They did not complete to correct yet. They should correct all.

Response: We apologize for omitting these important points. Efforts have been made to correct the misinformation. We hope our current revised manuscript meets the publication criteria in PlosOne.

---

## [Decision Letter · Decision Letter 2]

28 Nov 2022

PONE-D-22-03287R2A genome-wide analysis of the ASYMMETRIC LEAVES2-LIKE/LATERAL ORGAN BOUNDARIES (ASL/LBD) transcription factors (TFs) in cassava ( Manihot esculenta ) and expression profile under biotic and abiotic stressPLOS ONE

Dear Dr. chen,

Thank you for submitting your manuscript to PLOS ONE. After careful consideration, we feel that it has merit but does not fully meet PLOS ONE’s publication criteria as it currently stands. Therefore, we invite you to submit a revised version of the manuscript that addresses the points raised during the review process.

We look forward to receiving your revised manuscript.

Kind regards,

Vibhav Gautam

Academic Editor

PLOS ONE

Journal Requirements:

Reviewers' comments:

Reviewer's Responses to Questions

**Comments to the Author**

1. If the authors have adequately addressed your comments raised in a previous round of review and you feel that this manuscript is now acceptable for publication, you may indicate that here to bypass the “Comments to the Author” section, enter your conflict of interest statement in the “Confidential to Editor” section, and submit your "Accept" recommendation.

Reviewer #1: (No Response)

Reviewer #3: (No Response)

Reviewer #4: All comments have been addressed

Reviewer #5: (No Response)

2. Is the manuscript technically sound, and do the data support the conclusions?

Reviewer #1: Partly

Reviewer #3: Yes

Reviewer #4: Yes

Reviewer #5: Yes

3. Has the statistical analysis been performed appropriately and rigorously? 

Reviewer #1: I Don't Know

Reviewer #3: No

Reviewer #4: Yes

Reviewer #5: Yes

4. Have the authors made all data underlying the findings in their manuscript fully available?

Reviewer #1: Yes

Reviewer #3: Yes

Reviewer #4: Yes

Reviewer #5: Yes

5. Is the manuscript presented in an intelligible fashion and written in standard English?

Reviewer #1: Yes

Reviewer #3: Yes

Reviewer #4: Yes

Reviewer #5: Yes

6. Review Comments to the Author

Reviewer #1: The authors have corrected almost all. However, there are still ìnsuffícient part,<comment>

All AtLBD should be written AtASL/LBD, for example,

AtAS2/LBD6, AtASL1/LBD36, AtASL2/LBD10, AtASL3/LBD25, AtASL4/LOB, AtASL5/LBD12, AtASL6/LBD4, AtASL7/LBD11, AtASL8/LBD1, AtASL9/LBD3, AtASL10/LBD13, AtASL11/LBD15, AtASL12/LBD21, AtASL13/LBD24, AtASL14/LBD23, AtASL15/LBD17, AtASL16/LBD29, and so on.

See figure 4 and 5 in Machida et al., 2022 (https://link.springer.com/article/10.1007/s10265-021-01349-6).

Nomenclature with ASLs is important. This is because nomenclature with ASLs is based on the amino acid sequence homology of the AS2/LOB domain. Similar amino acid sequences have been shown to be functionally similar. Also in cassava, it will be possible to infer their functions by comparing with ASLs of Arabidopsis thaliana.

Two types of nomenclature (ASLs and LBDs in Arabidopsis thaliana) should be listed side by side and such nomenclature with ASLs should be advantageous for discussing relationships between the phylogeny and developmental functions of members in the AS2/LOB protein family. For example, at least four known proteins involved in auxin-induced lateral root formation are named ASL18/LBD16, ASL16/LBD29, ASL20/LBD18, and ASL24/LBD33; ASL15/LBD17 is also proposed to play a similar role. All of them are located in the close clades of sub-class Class Ia of A. thaliana in the phylogenetic tree. During pollen development, ASL1/LBD36, ASL2/LBD10, and ASL3/LBD25, all of which belong to the close narrow clades of the tree, control asymmetric cell division during pollen development. Interestingly, another set of ASL members (SCP/ASL29/LBD27 and ASL30/LBD22 in a small clade) are also involved in the same single process, but at different steps during the progression of pollen development. Note that ASL1/LBD36, ASL2/LBD10, and ASL3/LBD25 belong to Class Ia of A. thaliana, and that SCP/ASL29/LBD27 and ASL30/LBD22 belong to Class Ib of A. thaliana. Use of the ASL nomenclature might provide an edge to discussion of evolutionary developmental biology (evo-devo) of the AS2/LOB family.</comment>

Reviewer #3: The manuscript (PONE-D-22-03287) is well-prepared and quite conclusive. The language used in the manuscript is appropriate. The observations have been presented in a proper scientific manner and have been discussed well. Still, there is a scope to uplift the MS, I have a couple of suggestive comments which should further improve the manuscript.

Major comments

[1] The leaf lesion experiment in the VIGS lines is an important experiment, and currently, the images of the leaves provided are not very convincing. I encourage the authors to provide an image captured under similar light conditions (currently, the brightness and contrast of all the leaf areas are different). Also, Mark the inoculation point with a marker so that the inoculation site and lesion will be visible. Provide a picture of the entire leaf rather than a small section, then zoom in on the necrotic lesion area.

[2] Line#367 is there any information about these cultivators' abiotic and biotic stress sensitivity/tolerance? If yes, then provide that will strengthen this motive of the experiment.

Minor comments

[1] The discussion does not contain the figure references while mentioning the result, and please mention the figure number in the discussion too.

[2] Mention the statistical test performed while analyzing the data to test the significance in Figures 5 and 12

[3] Figure 6- Provide an explanation in the legend about the color code provided in the figure.

[4] Figure7- Specify tissue type and the variety of cassava used for transcription analysis as mentioned in the figure

[5] Figure 8- explain why the MeASLBD19 expression profile is missing (indicated by the gray color) in the disease group.

[6] Figure 9- The graph seems confusing; in MeASLBD12 and a few others, the relative expression value is significantly high at six h, but it still shows ns. This needs correction. Mention the statistical test performed.

[7] Fig 12b – provide scale bar.

Line#42 Gene expression is not a good keyword. Replace it.

Line#55, "development of fat symmetric," is this fat or flat?

Line#93 what is the basis of dividing into two subclasses? Class II is also divided into subclasses (a-g) which need to be mentioned.

Line#151. mentions other cassava cultivars, Arg7, W14, and KU50, that were used for gene analysis.

Line#155 Please specify how many leaves were inoculated per plant.

Line#156- Write bacterial dilution correctly using the superscript.

Line#156-Keep consistency in writing optical density here, it is OD600, while in line#194, it is written as (OD600).

Line#157- replace "hole" with "ring"

Line#162 - why only two similar replicates were considered?

Line#190-192 this explanation is not required, abridged.

Line#235 -Provide the hyperlink for the source code and web page for MCscanX.

Line#238 -Provide the hyperlink of the source for the Circos software

Line#256 provide the source website for the HISAT2 software.

Line#378 - RE-frame this sentence as "These results suggest the genotype-dependent tissue expression of these genes." Authors are using genotype and varieties as synonyms which is not the case. For example, in Line#382," cassava cultivars (Arg7 and W14)", these are referred to as Cultivars. However, KU50 and Arg7 are cultivated varieties and W14 wild ancestors. Correct this throughout the MS.

Line #382-Why the Variety KU50 not included in the expression analysis under drought treatment studies?

Line #385 "had significant differential expression under drought stress." Mention the fold change in expression compared to the control group.

Line# 427- "The 347 genes could be divided into four modules via WGCNA" explain the basis of this division into four modules indicated by different colors.

Reviewer #4: I found the paper to be well-prepared, with excellent and careful data selection, and it addressed all of the previous reviewers' comments with evidence. This MS could be useful for learning more about how ASYMMETRIC LEAVES2-LIKE/LATERAL ORGAN BOUNDARIES (ASL/LBD) transcription factors (TFs) in plants regulates biotic and abiotic stress responses. Therefore, this MS will timely provide important insight to understand ASL/LBD genes in plants under biotic and abiotic stress responses. However, main concerns associated with the MS are as follows:

(I)The authors have corrected most of the things suggested by previous reviewers: 1 and 2. However, I suggest to author please double check the AtLBD. It should be written AtASL/LBD in whole MS including figures and figure legends as also suggested by previous reviewers (twice).

(II)The figures and texts of Figure 1, Figure 2, Figure 3, Figure 10, and Figure S1 are unclear. I suggest authors should include high-resolution figures in their work to make it more understandable to a wider audience.

Reviewer #5: Mao and Abdoulaye et al - Genome-wide characterization of cassava LATERAL ORGAN BOUNDARIES domain genes revealed LBD47 involved in bacterial blight defense – PLOS ONE

The paper builds the study about genome-wide characterization of LBD genes using HMM profiles while establishing the role of MeLBD47 in plant phytohormone signalling in cassava. It showed significantly mitigated virulence of cassava bacterial blight (Xam CHN11) through Virus-induced gene silencing (VIGS).

General comment: The manuscript has the topic on the whole well covered and contains interesting set of experiments including VIGS and bioinformatic analyses that can be suitable for publication. Here are my few concerns:

Title: Please reframe the title of the paper. It does not sound very conclusive. One suggestion is “Genome-wide analyses of LATERAL ORGAN BOUNDARIES in cassava reveal the role of LBD47 in defense against bacterial blight”.

Line 23: ‘T’ small case in Arabidopsis thaliana

Line 32: ‘MeLBD47 was selected…’. How was it selected? Randomly? If yes, please mention that and if no, mention what was the basis of its selection for functional analyses?

Line 32 and 33: MeLBD47 is italicised somewhere and somewhere not. Please maintain uniformity.

Line 65: Add space between esculentaCrantz

Line 72: ‘extand’ misspelled

Line 99: 2-ΔCT or 2-ΔΔCT ?

Line 197-201: Information is repeated from Introduction.

Line 283: ‘leaf’, not ‘leave’

Line 343: ‘inoculated with’, not ‘Inoculated’

Discussion: Authors have mentioned in the introduction that this information will provide invaluable insights for further studies. I would suggest that the paper will be more informative if few lines are added in the discussion part about how this information can be put in use and what kinds of further studies can be done.

7. PLOS authors have the option to publish the peer review history of their article (what does this mean?). If published, this will include your full peer review and any attached files.

Reviewer #1: No

Reviewer #3: No

Reviewer #4: No

Reviewer #5: No

---

## [Author Response · Author response to Decision Letter 2]

1 Feb 2023

Dr. Yinhua Chen

Hainan Key Laboratory for Sustainable Utilisation of Tropical Bioresources,

College of Tropical Crops,

Hainan University,

Haikou 570228, Hainan, China

16.01.2023

Dear Editor-in-Chief,

With great pleasure, I re-submit the revised version of the manuscript ID PONE-D-22-03287 entitled "Genome-wide characterisation of cassava LATERAL ORGAN BOUNDARIES domain genes revealed LBD47 involved in bacterial blight defense" for consideration by PlosOne. Firstly, I would like to thank you for the opportunity to revise this manuscript. I highly appreciated the constructive comments and fruitful suggestions provided by each reviewer. Notably, some comments stemmed from the first round revised manuscript, probably because two reviewers were assigned to the first revision. Therefore, some comments were already addressed previously. The manuscript has certainly benefited from these insightful suggestions. Therefore, I look forward to working with you and the reviewers to move this manuscript for publication in the journal of PlosOne.

We have responded to all the reviewer's comments. Major improvements were made accordingly.

Reviewer #1: The authors have corrected almost all. However, there are still ìnsuffícient part, 

All AtLBD should be written AtASL/LBD, for example,

AtAS2/LBD6, AtASL1/LBD36, AtASL2/LBD10, AtASL3/LBD25, AtASL4/LOB, AtASL5/LBD12, AtASL6/LBD4, AtASL7/LBD11, AtASL8/LBD1, AtASL9/LBD3, AtASL10/LBD13, AtASL11/LBD15, AtASL12/LBD21, AtASL13/LBD24, AtASL14/LBD23, AtASL15/LBD17, AtASL16/LBD29, and so on.

See figure 4 and 5 in Machida et al., 2022 (https://link.springer.com/article/10.1007/s10265-021-01349-6).

Nomenclature with ASLs is important. This is because nomenclature with ASLs is based on the amino acid sequence homology of the AS2/LOB domain. Similar amino acid sequences have been shown to be functionally similar. Also in cassava, it will be possible to infer their functions by comparing with ASLs of Arabidopsis thaliana.

Two types of nomenclature (ASLs and LBDs in Arabidopsis thaliana) should be listed side by side and such nomenclature with ASLs should be advantageous for discussing relationships between the phylogeny and developmental functions of members in the AS2/LOB protein family. For example, at least four known proteins involved in auxin-induced lateral root formation are named ASL18/LBD16, ASL16/LBD29, ASL20/LBD18, and ASL24/LBD33; ASL15/LBD17 is also proposed to play a similar role. All of them are located in the close clades of sub-class Class Ia of A. thaliana in the phylogenetic tree. During pollen development, ASL1/LBD36, ASL2/LBD10, and ASL3/LBD25, all of which belong to the close narrow clades of the tree, control asymmetric cell division during pollen development. Interestingly, another set of ASL members (SCP/ASL29/LBD27 and ASL30/LBD22 in a small clade) are also involved in the same single process, but at different steps during the progression of pollen development. Note that ASL1/LBD36, ASL2/LBD10, and ASL3/LBD25 belong to Class Ia of A. thaliana, and that SCP/ASL29/LBD27 and ASL30/LBD22 belong to Class Ib of A. thaliana. Use of the ASL nomenclature might provide an edge to discussion of evolutionary developmental biology (evo-devo) of the AS2/LOB family.

Response: Thanks for this great point. The two terminology types were listed side by side (ASLs and LBDs). Please see figure 1

Reviewer #3: The manuscript (PONE-D-22-03287) is well-prepared and quite conclusive. The language used in the manuscript is appropriate. The observations have been presented in a proper scientific manner and have been discussed well. Still, there is a scope to uplift the MS, I have a couple of suggestive comments which should further improve the manuscript.

Response: Thanks for taking the time to read our manuscript and for all the constructive comments.

Major comments

[1] The leaf lesion experiment in the VIGS lines is an important experiment, and currently, the images of the leaves provided are not very convincing. I encourage the authors to provide an image captured under similar light conditions (currently, the brightness and contrast of all the leaf areas are different). Also, Mark the inoculation point with a marker so that the inoculation site and lesion will be visible. Provide a picture of the entire leaf rather than a small section, then zoom in on the necrotic lesion area.

Response: We apologize for the light conditions of the pictures. We have improved the picture with a scale bar, and hopefully, it is convincing enough. Please see picture 12

[2] Line#367 is there any information about these cultivators' abiotic and biotic stress sensitivity/tolerance? If yes, then provide that will strengthen this motive of the experiment.

Response: Thanks for these suggestions. Indeed, it could improve our manuscript. However, we haven’t conducted these analyses. And we apologize that we couldn’t conduct these analyses due to the time given.

Minor comments

[1] The discussion does not contain the figure references while mentioning the result, and please mention the figure number in the discussion too.

Response: Thanks for pointing these out. We have now referenced the figures in the revised manuscript. Please see the discussion section.

[2] Mention the statistical test performed while analyzing the data to test the significance in Figures 5 and 12

Response: We have now mentioned the statistical test performed. Please see figures 5 and 12

[3] Figure 6- Provide an explanation in the legend about the color code provided in the figure.

Response: We have now added an explanation about the color bar. Please see figure 6 legend

[4] Figure7- Specify tissue type and the variety of cassava used for transcription analysis as mentioned in the figure

Response: We Have now specified the tissue type and cassava varieties. Please see the figure 7 legend.

[5] Figure 8- explain why the MeASLBD19 expression profile is missing (indicated by the gray color) in the disease group.

Response: When we downloaded the expression profile data of cassava LBDs in diseases, we found that the expression abundance of MeASLBD19 was 0, therefore, it was displayed in gray.

[6] Figure 9- The graph seems confusing; in MeASLBD12 and a few others, the relative expression value is significantly high at six h, but it still shows ns. This needs correction. Mention the statistical test performed.

Response: Thanks for pointing out these typos. It has now been revised. Please see figure 9

[7] Fig 12b – provide scale bar.

Response: Thanks; we have provided a scale bar. Please see figure 12b

Line#42 Gene expression is not a good keyword. Replace it.

Response: Thanks. Gene expression has been replaced with expression profiles. Please see line 41 

Line#55, "development of fat symmetric," is this fat or flat?

Response: Thanks for pointing out this typo. It has been corrected. Please see line 54

Line#93 what is the basis of dividing into two subclasses? Class II is also divided into subclasses (a-g) which need to be mentioned.

Response: The tree was divided into two classes based on the clades form and previous studies that divided the arabidopsis LOD proteins into two classes: https://academic.oup.com/plphys/article/129/2/747/6110259#265176879

However, the subclasses (a-g) could not be mentioned because they resulted from the current studies. 

Line#151. mentions other cassava cultivars, Arg7, W14, and KU50, that were used for gene analysis.

Response: Thanks for pointing it out. We have mentioned the missed cultivars. Please see 150

Line#155 Please specify how many leaves were inoculated per plant.

Response: It has been mentioned. Please see line 154

Line#156- Write bacterial dilution correctly using the superscript.

Response: Thanks for pointing it out. It has been corrected. Please see lines 155

Line#156-Keep consistency in writing optical density here, it is OD600, while in line#194, it is written as (OD600).

Response: We apologised for this inconsistency. We have corrected it with OD600 throughout the revised manuscript.

Line#157- replace "hole" with "ring"

Response: Thanks. We have replaced hole with ring. Please see line 156

Line#162 - why only two similar replicates were considered?

Response: We meant, “The experiments were conducted at least twice with identical results”.

Line#190-192 this explanation is not required, abridged.

Response: We have abridged the sentence. Please see line 138

Line#235 -Provide the hyperlink for the source code and web page for MCscanX.

Response: Thanks. The source code and web page have been provided. Please see line 235

Line#238 -Provide the hyperlink of the source for the Circos software

Response: Thanks. The hyperlink of the source for the circus software has been provided. Please see line 238

Line#256 provide the source website for the HISAT2 software.

Response: Thanks. The source website for the HISAT2 software has been provided. Please see line 257

Line#378 - RE-frame this sentence as "These results suggest the genotype-dependent tissue expression of these genes." Authors are using genotype and varieties as synonyms which is not the case. For example, in Line#382," cassava cultivars (Arg7 and W14)", these are referred to as Cultivars. However, KU50 and Arg7 are cultivated varieties and W14 wild ancestors. Correct this throughout the MS.

Response: Thanks for pointing these out, and we apologise for the misleading. We have reframed the sentence. Please see lines 378-379

Besides, we have corrected these misleading sentences throughout the revised manuscript.

Line #382-Why the Variety KU50 not included in the expression analysis under drought treatment studies?

Response: Because the expression profiling data (downloaded from NCBI) did not contain the expression pattern of KU50 under drought.

Line #385 "had significant differential expression under drought stress." Mention the fold change in expression compared to the control group.

Response: Thanks for your great comment. Since the folds were not mentioned on the heat map, we think it might be better not to stay consistent with figure 8. 

Line# 427- "The 347 genes could be divided into four modules via WGCNA" explain the basis of this division into four modules indicated by different colors.

Response: Highly interconnected gene sets were obtained by WGNCA analysis, and these gene sets were designated as "modules". A total of 4 modules were obtained. Among them, the gray module was considered a non-coexpressed gene set by default; therefore, it was not considered in the analysis. Please see lines 428-429

Reviewer #4: I found the paper to be well-prepared, with excellent and careful data selection, and it addressed all of the previous reviewers' comments with evidence. This MS could be useful for learning more about how ASYMMETRIC LEAVES2-LIKE/LATERAL ORGAN BOUNDARIES (ASL/LBD) transcription factors (TFs) in plants regulates biotic and abiotic stress responses. Therefore, this MS will timely provide important insight to understand ASL/LBD genes in plants under biotic and abiotic stress responses. However, main concerns associated with the MS are as follows:

(I)The authors have corrected most of the things suggested by previous reviewers: 1 and 2. However, I suggest to author please double check the AtLBD. It should be written AtASL/LBD in whole MS including figures and figure legends as also suggested by previous reviewers (twice).

Response: Thanks for taking the time to read our manuscript and for all your suggestions. The two terminology types were listed side by side (ASLs and LBDs) throughout the revised manuscript.

(II)The figures and texts of Figure 1, Figure 2, Figure 3, Figure 10, and Figure S1 are unclear. I suggest authors should include high-resolution figures in their work to make it more understandable to a wider audience.

Response: We apologise for the unclear pictures. We have replaced them with high-resolution throughout the revised manuscript.

Reviewer #5: Mao and Abdoulaye et al - Genome-wide characterization of cassava LATERAL ORGAN BOUNDARIES domain genes revealed LBD47 involved in bacterial blight defense – PLOS ONE

The paper builds the study about genome-wide characterization of LBD genes using HMM profiles while establishing the role of MeLBD47 in plant phytohormone signalling in cassava. It showed significantly mitigated virulence of cassava bacterial blight (Xam CHN11) through Virus-induced gene silencing (VIGS).

General comment: The manuscript has the topic on the whole well covered and contains interesting set of experiments including VIGS and bioinformatic analyses that can be suitable for publication. Here are my few concerns:

Title: Please reframe the title of the paper. It does not sound very conclusive. One suggestion is “Genome-wide analyses of LATERAL ORGAN BOUNDARIES in cassava reveal the role of LBD47 in defense against bacterial blight”.

Response: Thanks for your comments and time to read our manuscript. The title has been reframed with the suggested title. Nevertheless, we would like to mention that most of the comments ref (Lines) were based on the first revised manuscript. Therefore we have already revised some parts previously.

Line 23: ‘T’ small case in Arabidopsis thaliana

Response: Thanks for pointing out this typo. It has been corrected. Please see line 21

Line 32: ‘MeLBD47 was selected…’. How was it selected? Randomly? If yes, please mention that and if no, mention what was the basis of its selection for functional analyses?

Response: MeLBD47 was selected based on the expression level under disease treatment and drought stress.

Line 32 and 33: MeLBD47 is italicised somewhere and somewhere not. Please maintain uniformity.

Response: We apologise for the inconsistency. We have revised it throughout the revised manuscript.

Line 65: Add space between esculentaCrantz

Response: It has been corrected

Line 72: ‘extand’ misspelled

Response: It has been corrected

Line 99: 2-ΔCT or 2-ΔΔCT ?

Response: It has been corrected

Line 197-201: Information is repeated from Introduction.

Response: It has been revised.

Line 283: ‘leaf’, not ‘leave’

Response: Thanks for pointing out these typos. It has been corrected throughout the revised manuscript.

Line 343: ‘inoculated with’, not ‘Inoculated’

Response: Thanks for pointing out these typos. It has been corrected throughout the revised manuscript.

Discussion: Authors have mentioned in the introduction that this information will provide invaluable insights for further studies. I would suggest that the paper will be more informative if few lines are added in the discussion part about how this information can be put in use and what kinds of further studies can be done.

Response: Thanks for these great suggestions. We have revised the discussion section accordingly, and hopefully, it meets the requirements.

---

## [Editor Report · Decision Letter 3]

8 Feb 2023

Genome-wide analyses of LATERAL ORGAN BOUNDARIES in cassava reveal the role of LBD47 in defence against bacterial blight

PONE-D-22-03287R3

Dear Dr. chen,

We’re pleased to inform you that your manuscript has been judged scientifically suitable for publication and will be formally accepted for publication once it meets all outstanding technical requirements.

Kind regards,

Vibhav Gautam

Academic Editor

PLOS ONE
---

## [Editor Report · Acceptance letter]

10 Apr 2023

PONE-D-22-03287R3 

Genome-wide analyses of LATERAL ORGAN BOUNDARIES in cassava reveal the role of LBD47 in defence against bacterial blight 

Dear Dr. Chen:

I'm pleased to inform you that your manuscript has been deemed suitable for publication in PLOS ONE. Congratulations! Your manuscript is now with our production department. 

Kind regards, 

on behalf of

Dr. Vibhav Gautam 

Academic Editor

PLOS ONE